# IMPROVING GRAPH NEURAL NETWORK EXPRESSIVITY VIA SUBGRAPH ISOMORPHISM COUNTING

## ABSTRACT

While Graph Neural Networks (GNNs) have achieved remarkable results in a variety of applications, recent studies exposed important shortcomings in their ability to capture the structure of the underlying graph. It has been shown that the expressive power of standard GNNs is bounded by the Weisfeiler-Lehman (WL) graph isomorphism test, from which they inherit proven limitations such as the inability to detect and count graph substructures. On the other hand, there is significant empirical evidence, e.g. in network science and bioinformatics, that substructures are often informative for downstream tasks, suggesting that it is desirable to design GNNs capable of leveraging this important source of information. To this end, we propose a novel topologically-aware message passing scheme based on substructure encoding. We show that our architecture allows incorporating domain-specific inductive biases and that it is strictly more expressive than the WL test. Importantly, in contrast to recent works on the expressivity of GNNs, we do not attempt to adhere to the WL hierarchy; this allows us to retain multiple attractive properties of standard GNNs such as locality and linear network complexity, while being able to disambiguate even hard instances of graph isomorphism. We extensively evaluate our method on graph classification and regression tasks and show state-of-the-art results on multiple datasets including molecular graphs and social networks.

## 1 INTRODUCTION

The field of graph representation learning has undergone a rapid growth in the past few years. In particular, Graph Neural Networks (GNNs), a family of neural architectures designed for irregularly structured data, have been successfully applied to problems ranging from social networks and recommender systems (Ying et al., 2018a) to bioinformatics (Fout et al., 2017; Gainza et al., 2020), chemistry (Duvenaud et al., 2015; Gilmer et al., 2017; Sanchez-Lengeling et al., 2019) and physics (Kipf et al., 2018; Battaglia et al., 2016), to name a few. Most GNN architectures are based on message passing (Gilmer et al., 2017), where at each layer the nodes are updated by information aggregated from their neighbours.

A crucial difference from traditional neural networks operating on grid-structured data is the absence of canonical ordering of the nodes in a graph. To address this, the aggregation function is constructed to be invariant to neighbourhood permutations and, as a consequence, to graph isomorphism. This kind of symmetry is not always desirable and thus different inductive biases that disambiguate the neighbours have been proposed. For instance, in geometric graphs, such as 3D molecular graphs and meshes, directional biases are usually employed in order to model the positional information of the nodes (Masci et al., 2015; Monti et al., 2017; Bouritsas et al., 2019; Klicpera et al., 2020; de Haan et al., 2020b); for proteins, ordering information is used to disambiguate amino-acids at different positions in the sequence (Ingraham et al., 2019); in multi-relational knowledge graphs, a different aggregation is performed for each relation type (Schlichtkrull et al., 2018).

The structure of the graph itself does not usually explicitly take part in the aggregation function. In fact, most models rely on multiple message passing steps as a means for each node to discover the global structure of the graph. However, since message-passing GNNs are at most as powerful as the Weisfeiler Lehman test (WL) (Xu et al., 2019; Morris et al., 2019), they are limited in their abilities to adequately exploit the graph structure, e.g. by counting substructures (Arvind et al., 2019;

Chen et al., 2020). This uncovers a crucial limitation of GNNs, as substructures have been widely recognised as important in the study of complex networks. For example, in molecular chemistry, functional groups and rings are related to a plethora of chemical properties, while cliques are related to protein complexes in Protein-Protein Interaction networks and community structure in social networks, respectively (Granovetter, 1982; Girvan & Newman, 2002).

Motivated by this observation, in this work we propose *Graph Substructure Network (GSN)*, a new symmetry breaking mechanism for GNNs based on introducing structural biases in the aggregation function. In particular, each message is transformed differently depending on the topological relationship between the endpoint nodes. This relationship is expressed by counting the appearance of certain substructures, the choice of which allows us to provide the model with different inductive biases, based on the graph distribution at hand. The substructures are encoded as structural identifiers that are assigned to either the nodes or edges of the graph and can thus disambiguate the neighbouring nodes that take part in the aggregation.

We characterise the expressivity of substructure encoding in GNNs, showing that GSNs are strictly more expressive than traditional GNNs for the vast majority of substructures while retaining the locality of message passing, as opposed to higher-order methods (Maron et al., 2019b;c;a; Morris et al., 2019) that follow the WL hierarchy (see Section 2). In the limit, our model can yield a unique representation for every isomorphism class and is thus universal. We provide an extensive experimental evaluation on hard instances of graph isomorphism testing (strongly regular graphs), as well as on real-world networks from the social and biological domains, including the recently introduced large-scale benchmarks (Dwivedi et al., 2020; Hu et al., 2020). We observe that when choosing the structural inductive biases based on domain-specific knowledge, GSN achieves state-of-the-art results.

## 2 PRELIMINARIES

Let $G = (\mathcal{V}_G, \mathcal{E}_G)$ be a graph with vertex set $\mathcal{V}_G$ and undirected edge set $\mathcal{E}_G$. A subgraph $G_S = (\mathcal{V}_{G_S}, \mathcal{E}_{G_S})$ of $G$ is any graph with $\mathcal{V}_{G_S} \subseteq \mathcal{V}_G$, $\mathcal{E}_{G_S} \subseteq \mathcal{E}_G$. When $\mathcal{E}_{G_S}$ includes all the edges of $G$ with endpoints in $\mathcal{V}_{G_S}$, i.e. $\mathcal{E}_{G_S} = \{(v, u) \in \mathcal{E} : v, u \in \mathcal{V}_{G_S}\}$, the subgraph is said to be *induced*.

**Isomorphisms** Two graphs $G, H$ are *isomorphic* (denoted $H \simeq G$), if there exists an adjacency-preserving bijective mapping (*isomorphism*) $f : \mathcal{V}_G \to \mathcal{V}_H$, i.e. $(v, u) \in \mathcal{E}_G$ iff $(f(v), f(u)) \in \mathcal{E}_H$. Given some small graph $H$, the *subgraph isomorphism* problem amounts to finding a subgraph $G_S$ of $G$ such that $G_S \simeq H$. An *automorphism* of $H$ is an isomorphism that maps $H$ onto itself. The set of all the unique automorphisms form the *automorphism group* of the graph, denoted as $\mathrm{Aut}(H)$, contains all the possible symmetries of the graph. The automorphism group yields a partition of the vertices into disjoint subsets of $\mathcal{V}_H$ called *orbits*. Intuitively, this concept allows us to group the vertices based on their *structural roles*, e.g. the end vertices of a path, or all the vertices of a cycle (see Figure 1). Formally, the orbit of a vertex $v \in \mathcal{V}_H$ is the set of vertices to which it can be mapped via an automorphism: $\mathrm{Orb}(v) = \{u \in \mathcal{V}_H : \exists g \in \mathrm{Aut}(H) \text{ s.t. } g(u) = v\}$, and the set of all orbits $H \setminus \mathrm{Aut}(H) = \{\mathrm{Orb}(v) : v \in \mathcal{V}_H\}$ is usually called the *quotient* of the automorphism when it acts on the graph $H$. We are interested in the unique elements of this set that we will denote as $\{O_{H,1}^V, O_{H,2}^V, \ldots, O_{H,d_H}^V\}$, where $d_H$ is the cardinality of the quotient.

Analogously, we define edge structural roles via *edge automorphisms*, i.e. bijective mappings from the edge set onto itself, that preserve edge adjacency (two edges are adjacent if they share a common endpoint). In particular, every vertex automorphism $g$ induces an edge automorphism by mapping each edge $\{u, v\}$ to $\{g(u), g(v)\}$. [1] In the same way as before, we construct the edge automorphism group, from which we deduce the partition of the edge set in *edge orbits* $\{O_{H,1}^E, O_{H,2}^E, \ldots, O_{H,d_H}^E\}$.

**Weisfeiler-Lehman tests:** The *Weisfeiler-Lehman graph-isomorphism test* (Weisfeiler & Leman, 1968), also known as naive vertex refinement, *1-WL*, or just *WL*), is a fast heuristic to decide if two graphs are isomorphic or not. The WL test proceeds as follows: every vertex $v$ is initially assigned a colour $c^0(v)$ that is later iteratively refined by aggregating neighbouring information:

---

[1]Note that the edge automorphism group is larger than that of induced automorphisms, but strictly larger only for 3 trivial cases (Whitney, 1932). However, induced automorphisms provide a more natural way to express edge structural roles.

$c^{t+1}(v) = \text{HASH}\Big(c^t(v),\ \wr c^t(u) \wr_{u \in \mathcal{N}(v)}\Big)$, where $\wr \cdot \wr$ denotes a multiset (a set that allows element repetitions) and $\mathcal{N}(v)$ is the neighbourhood of $v$. The WL algorithm terminates when the colours stop changing, and outputs a histogram of colours. Two graphs with different histograms are non-isomorphic; if the histograms are identical, the graphs are possibly, but not necessarily, isomorphic. Note that the neighbour aggregation in the WL test is a form of message passing, and GNNs are the learnable analogue.

A series of works on improving GNN expressivity mimic the higher-order generalisations of WL, known as $k$-WL and $k$-Folklore WL (WL hierarchy) and operate on $k$-tuples of nodes (see Appendix B.1). The $(k+1)$-FWL is strictly stronger than $k$-FWL, $k$-FWL is as strong as $(k+1)$-WL and 2-FWL is strictly stronger than the simple 1-WL test.

## 3    GRAPH SUBSTRUCTURE NETWORKS

Graphs consist of nodes (or edges) with repeated structural roles. Thus, it is natural for a neural network to treat them in a similar manner, akin to weight sharing between local patches in CNNs for images (LeCun et al., 1989) or positional encodings in language models for sequential data (Sukhbaatar et al., 2015; Gehring et al., 2017; Vaswani et al., 2017). Nevertheless, GNNs are usually unaware of the nodes' different structural roles, since all nodes are treated equally when performing local operations. Despite the initial intuition that the neural network would be able to discover these roles by constructing deeper architectures, it has been shown that GNNs are ill-suited for this purpose and are blind to the existence of structural properties, e.g. triangles or larger cycles (Chen et al., 2020; Arvind et al., 2019).

To this end, we propose to explicitly encode structural roles as part of message passing, in order to capture richer topological properties. Our method draws inspiration from Loukas (2020), where it was shown that GNNs become universal when the nodes in the graph are uniquely identified, i.e when they are equipped with different features. However, it is not clear how to choose these identifiers in a way that can allow the neural network to generalise. Structural roles, when treated as identifiers, although not necessarily unique, are more amenable to generalisation due to their repetition across different graphs. Thus, they can constitute a trade-off between uniqueness and generalisation.

**Structural features:** Structural roles are encoded into features by counting the appearance of certain substructures. The larger the substructure collection, the more fine-grained the partition of nodes into roles will be. Let $\mathcal{H} = \{H_1, H_2 \dots H_K\}$ be a set of small (connected) graphs, for example cycles of fixed length or cliques. For each graph $H \in \mathcal{H}$, we first find its isomorphic subgraphs in $G$ denoted as $G_S$. For each node $v \in \mathcal{V}_{G_S}$ we infer its role w.r.t. $H$ by obtaining the orbit of its mapping $f(v)$ in $H$, $\text{Orb}_H(f(v))$. By counting all the possible appearances of different orbits in $v$, we obtain the *vertex structural feature* $\mathbf{x}_H^V(v)$ of $v$, defined as follows:

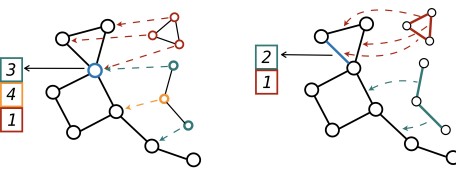

Figure 1: *Node* (left) and *edge* (right) induced subgraph counting for a 3-cycle and a 3-path. Counts are reported for the blue node on the left and for the blue edge on the right. Different colors depict orbits.

$$x_{H,i}^V(v) = \frac{|\{G_S \simeq H\ :\ v \in \mathcal{V}_{G_S}\ \text{s.t.}\ f(v) \in O_{H,i}^V\}|}{|\text{Aut}(H)|}, \quad i = 1, \dots, d_H. \tag{1}$$

We divide the counts by the number of the automorphisms of $H$, since for every matched subgraph $G_S$ there will always be $|\text{Aut}(H)|$ different ways to map it to $H$, thus these repetitions will be uninformative. By combining the counts from different substructures in $\mathcal{H}$ and different orbits, we obtain the feature vector $\mathbf{x}_V(v) = [\mathbf{x}_{H_1}^V(v), \dots, \mathbf{x}_{H_K}^V(v)] \in \mathbb{N}^{D \times 1}$ of dimension $D = \sum_{H_i \in \mathcal{H}} d_{H_i}$.

Similarly, we can define *edge structural features* $\mathbf{x}_{H,i}^E(\{u, v\})$ by counting occurrences of edge automorphism orbits:

$$x_{H,i}^E(\{u, v\}) = \frac{|\{G_S \simeq H\ :\ \{u, v\} \in \mathcal{E}_{G_S}\ \text{s.t.}\ \{f(u), f(v)\} \in O_{H,i}^E\}|}{|\text{Aut}(H)|}, \tag{2}$$

and the combined edge features $\mathbf{x}_E(\{u, v\}) = [\mathbf{x}_{H_1}^E(\{u, v\}), \dots, \mathbf{x}_{H_K}^E(\{u, v\})]$. An example of vertex and edge structural features is illustrated in Figure 1.

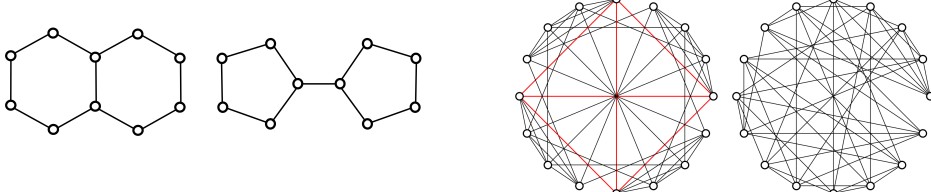

Figure 2: (Left) *Decalin* and *Bicyclopentyl*: Non-isomorphic molecular graphs than can be distinguished by GSN, but not the by the WL test (Sato, 2020) (nodes represent carbon atoms and edges chemical bonds). (Right) *Rook's 4x4 graph* and the *Shrikhande graph*: the smallest pair of strongly regular non-isomorphic graphs with the same parameters SR(16,6,2,2). GSN can distinguish them with 4-clique counts, while 2-FWL fails.

**Structure-aware message passing:** The key building block of our architecture is the graph substructure layer, defined in a general manner as a Message Passing Neural Network (MPNN) (Gilmer et al., 2017), where now the messages from the neighbouring nodes also contain the structural information. In particular, each node $v$ updates its state $\mathbf{h}^t(v)$ by combining its previous state with the aggregated messages: $\mathbf{h}^{t+1}(v) = \mathrm{UP}^{t+1}\big(\mathbf{h}^t(v), \mathrm{MSG}^{t+1}(v)\big)$, where the $\mathrm{UP}^{t+1}$ function is a multilayer perceptron (MLP) and the message aggregation is a summation of features transformed by an MLP $M^{t+1}$ as follows:

$$\mathrm{MSG}^{t+1}(v) \quad = \quad \sum_{u \in \mathcal{N}(v)} M^{t+1}\big(\mathbf{h}^t(v), \mathbf{h}^t(u), \mathbf{x}_V(v), \mathbf{x}_V(u), \mathbf{e}(\{u, v\})\big) \qquad (3)$$

$$\mathrm{MSG}^{t+1}(v) \quad = \quad \sum_{u \in \mathcal{N}(v)} M^{t+1}\big(\mathbf{h}^t(v), \mathbf{h}^t(u), \mathbf{x}_E(\{u, v\}), \mathbf{e}(\{u, v\})\big) \qquad (4)$$

where the two variants, named *GSN-v* and *GSN-e*, correspond to vertex- or edge-counts, respectively, and $\mathbf{e}(\{u, v\})$ denotes edge features. The two variants are analogous to absolute and relative positional encodings in language models (Shaw et al., 2018; Dai et al., 2019).

It is important to note here that contrary to identifier-based GNNs (Loukas, 2020; Sato et al., 2020; Dasoulas et al., 2020) that obtain universality at the expense of permutation equivariance (since the identifiers are arbitrarily chosen with the sole requirement of being unique), GSNs retain the attractive inductive bias of the permutation equivariant property. This stems from the fact that the process generating our structural identifiers (i.e. subgraph isomorphism) is permutation equivariant itself (proof provided in the appendix).

**How powerful are GSNs?** We now turn to the expressive power of GSNs in comparison to MPNNs and the WL tests, a key tool for the theoretical analysis of the expressivity of graph neural networks so far. Since GSN is a generalisation of MPNNs, it is easy to see that it is at least as powerful. Importantly, GSNs have the capacity to learn functions that traditional MPNNs cannot learn. The following observation derives directly from the analysis of the counting abilities of the 1-WL test (Arvind et al., 2019) and its extension to MPNNs (Chen et al., 2020) (for proofs, see Appendix A.2).

**Proposition 3.1.** *GSN is strictly more powerful than MPNN and the 1-WL test when one of the following holds: a) H is any graph except for star graphs of any size and structural features are inferred by subgraph matching, i.e. we count all subgraphs $G_S \cong H$ for which it holds that $\mathcal{E}_{G_S} \subseteq \mathcal{E}_G$, or b) H is any graph except for single edges and single nodes and structural features are inferred by **induced** subgraph matching, i.e. we count all subgraphs $G_S \cong H$ for which it holds that $\mathcal{E}_{G_S} = \mathcal{E}_G \cap \big(\mathcal{V}_{G_S} \times \mathcal{V}_{G_S}\big).$*

An important question is which substructures are the most *informative* and whether they can completely characterise the graph. As of today, we are not aware of any results in graph theory that can guarantee the reconstruction of a graph from a smaller collection of its subgraphs. In fact, the *Reconstruction Conjecture* (Kelly et al., 1957; Ulam, 1960), states that a graph with size $n \geq 3$ can be reconstructed from its vertex-deleted subgraphs. Moreover, it is known (Dasoulas et al., 2020; Chen et al., 2019) that solving graph isomorphism is equivalent to universal approximation of permutation invariant functions defined on graphs. Thus, we can state the following for GSN:

**Corollary 3.1.** *If the Reconstruction Conjecture holds, GSN can distinguish all non-isomorphic graphs of size $n$ and is therefore universal, when $\mathcal{H}$ contains all substructures of size $k = n - 1$.*

**How to choose the substructures?** The Reconstruction Conjecture has only been proven for $n \leq 11$ (McKay, 1997) and still remains open for larger graphs, while to the best of our knowledge, there is no similar hypothesis for smaller values of $k$. However, in practice we observe that small substructures of size $k = \mathcal{O}(1)$ are sufficient both for resolving hard instances of graph isomorphism as well as for tasks related to real-world networks. In particular, although our method does not attempt to align with the WL hierarchy, an interesting observation that we make is that small substructures have the capacity to distinguish *strongly regular* graphs, where 2-FWL provably fails (proven in Appendix B.3). This is illustrated in Figure 2 (right), where counting 4-cliques is sufficient to disambiguate this exemplary pair, and in Section 5, where a large scale experimental study is conducted.

In real-world scenarios, using too many and too large subgraphs will not only be expensive to compute, but they might also lead to overfitting. As an alternative, one can choose only the most discriminative subgraphs, i.e. the ones that can achieve the maximum possible node disambiguation, similarly to identifier-based approaches. Note that this approach solely takes into account expressivity and does not provide any guarantee about generalisation. This is empirically validated in Figure 4, where we observe that choosing substructures with strong node disambiguation properties allows for better fitting of the training data, but the test set performance does not necessarily improve.

Aiming at better generalisation, it is desirable to make use of substructures for which there is substantial prior knowledge of their importance in certain network distributions and have been observed to be intimately related to various properties. For example, small substructures (graphlets) have been extensively analysed in protein-protein interaction networks (Pržulj et al., 2004), triangles and cliques characterise the structure of ego-nets and social networks in general (Granovetter, 1982), simple cycles (rings) are central in molecular distributions, directed and temporal motifs have been shown to explain the working mechanisms of gene regulatory networks, biological neural networks, transportation networks and food webs (Milo et al., 2002; Paranjape et al., 2017; Benson et al., 2016). In Figure 6 in the Appendix, we showcase the importance of these inductive biases: a cycle-based GSN predicting molecular properties achieves smaller generalisation gap compared to a traditional MPNN, while at the same time generalising better with less training data. Choosing the best substructure collection is still an open problem; various heuristics can be used (motif frequencies, feature selection strategies) or ideally by learning the substructure dictionary in an end-to-end manner. Answering this question is left for future work.

**GSN-v vs GSN-e**: Finally, we examine the expressive power of the two proposed variants. A crucial observation that we make, is that for each graph $H$ in the collection, the vertex structural identifiers can be reconstructed by the corresponding edge identifiers. Thus, we can show that for every GSN-v there exists a GSN-e that can simulate the behaviour of the former (proof provided in Appendix A.3).

**Proposition 3.2.** *For a given subgraph collection $\mathcal{H}$, let $C^V$ the set of functions that can be expressed by a GSN-v with arbitrary depth and with, and $C^E$ the set of functions that can be expressed by a GSN-e with the same properties. Then, it holds that $C^E \supseteq C^V$, or in other words GSN-e is at least as expressive as GSN-v.*

**Complexity:** The complexity of GSN comprises two parts: precomputation (substructure counting) and training/testing. The key appealing property is that training and inference are linear w.r.t the number of edges, $\mathcal{O}(|\mathcal{E}|)$, as opposed to higher-order methods (Maron et al., 2019a; Morris et al., 2019) with $\mathcal{O}(n^k)$ training complexity and relational pooling (Murphy et al., 2019) with $\mathcal{O}(n!)$ training complexity in absence of approximations.

The worst-case complexity of subgraph isomorphism of fixed size $k$ is $\mathcal{O}(n^k)$, by examining all the possible $k$-tuples in the graph. However, for specific types of subgraphs, such as paths and cycles, the problem can be solved even faster (see e.g. Giscard et al. (2019)). Approximate counting algorithms are also widely used, especially for counting frequent network motifs (Kashtan et al., 2004; Wernicke, 2005; 2006; Wernicke & Rasche, 2006), and can provide a considerable speed-up. Furthermore, recent neural approaches (Ying et al., 2020b;a) provide fast approximate counting.

In our experiments, we performed exact counting using the common isomorphism algorithm VF2 (Cordella et al., 2004). Although its worst case complexity is $\mathcal{O}(n^k)$, it scales better in practice, for instance when the candidate subgraph is infrequently matched or when the graphs are sparse, and is also trivially parallelisable. In Figure 3, we show a quantitative analysis of the empirical runtime of the counting algorithm against the worst case, for three different graph distributions: molecules,

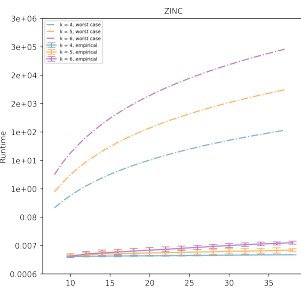 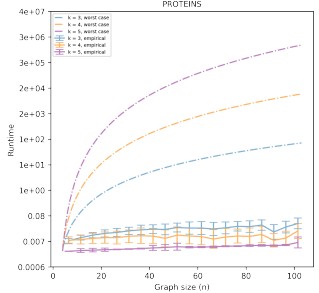 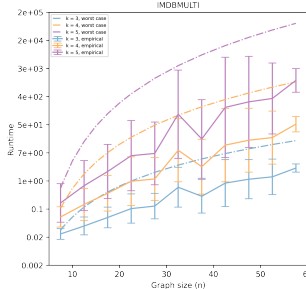

Figure 3: Empirical (solid) vs worst case (dashed) runtime for different graph distributions (in seconds, log scale). For each distribution we count the best performing (and frequent) substructures of increasing sizes $k$.

protein contact maps, social networks. It is easy to see that when the graphs are sparse (for the first two cases) and the number of matches is small, the algorithm is significantly faster than the worst case, while it scales better with the size of the graph $n$. Even, in the case of social networks, where several examples are near-complete graphs, both the runtime and the growth w.r.t both $n$ and $k$ are better than the worst case. Overall, the preprocessing computational burden in most of the cases remains negligible for relatively small and sparse graphs, as it is the case of molecules.

## 4 RELATED WORK

**Expressive power of GNNs and the WL hierarchy**: The seminal results in the theoretical analysis of the expressivity of GNNs (Xu et al., 2019) and k-GNNs (Morris et al., 2019) established that traditional message passing-based GNNs are at most as powerful as the 1-WL test. Chen et al. (2019) showed that graph isomorphism is equivalent to universal invariant function approximation. Higher-order Invariant Graph Networks (IGNs) have been studied in a series of works (Maron et al., 2019b;c;a; Chen et al., 2019; Keriven & Peyré, 2019; Puny et al., 2020), establishing connections with the WL hierarchy, similarly to Morris et al. (2019; 2020). The main drawback of these methods is the training and inference time complexity and memory requirements of $\mathcal{O}(n^k)$ as well as the super-exponential number of parameters for linear IGNs making them impractical.

From a different perspective, Sato et al. (2019) and Loukas (2020) showed the connections between GNNs and distributed local algorithms (Angluin, 1980; Linial, 1992; Naor & Stockmeyer, 1993) and suggested more powerful alternatives based on either local orderings or unique global identifiers (in the form of random features in Sato et al. (2020)) that make GNNs universal. However, these methods lack a principled way to choose orderings/identifiers that can be shared across graphs (this would require a graph canonisation procedure). Garg et al. (2020) also analysed the expressive power of MPNNs and other more powerful variants and provided generalisation bounds. Murphy et al. (2019) take into account all possible node permutations and can therefore be intractable. Concurrently with our work, more expressive GNNs have been proposed using equivariant message passing with different kernels for each edge based on their isomorphism classes (de Haan et al., 2020a), message passing with matrices of order equal to the size of the graph instead of vectors (Vignac et al., 2020), or by enhancing node features with distance encodings (Li et al., 2020).

Solely quantifying the expressive power of GNNs in terms of their ability to distinguish non-isomorphic graphs does not provide the necessary granularity: even the 1-WL test can distinguish almost all (in the probabilistic sense) non-isomorphic graphs (Babai et al., 1980). As a result, there have been several efforts to analyse the power of $k$-WL tests in comparison to other graph invariants (Fürer, 2010; 2017; Arvind et al., 2019; Dell et al., 2018), while recently Chen et al. (2020) approached GNN expressivity by studying their ability to count substructures.

**Substructures in Complex Networks:** The idea of analysing complex networks based on small-scale topological characteristics dates back to the 1970's and the notion of triad census for directed graphs (Holland & Leinhardt, 1976). The seminal paper of Milo et al. (2002) coined the term *network motifs* as over-represented subgraph patterns that were shown to characterise certain functional properties of complex networks in systems biology. The closely related concept of *graphlets* (Pržulj et al., 2004; Pržulj, 2007; Milenković & Pržulj, 2008; Sarajlić et al., 2016), different from motifs in

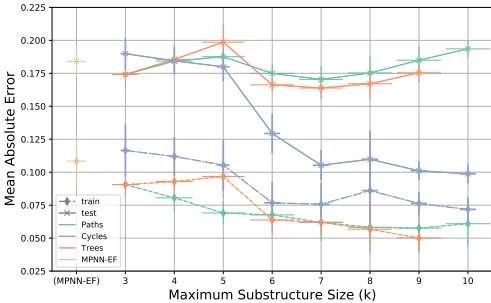 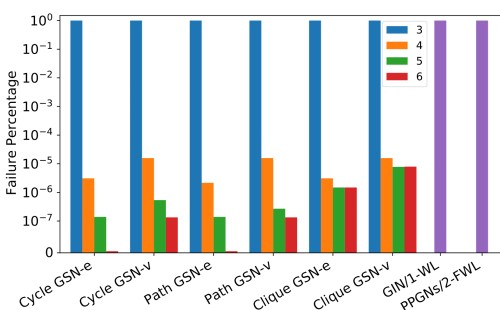

Figure 4: Train (dashed) and test (solid) MAEs for Path- and Cycle-GSN-*EF* as a function of the substructure maximum size $k$. Vertical bars indicate standard deviation; horizontal bars depict disambiguation/uniqueness scores $\delta$.

Figure 5: SR graphs isomorphism test (log scale, smaller values are better). Different colours indicate different substructure sizes.

being induced subgraphs, has been used to analyse the distribution of real-world networks and as a topological signature for network similarity. Our work is similar in spirit with the *graphlet degree vector* (GDV) (Pržulj, 2007), a node-wise descriptor based on graphlet counting.

Substructures have been also used in the context of ML. In particular, subgraph patterns have been used to define Graph Kernels (GKs) (Horváth et al., 2004; Shervashidze et al., 2009; Costa & De Grave, 2010; Kriege & Mutzel, 2012; NT & Maehara, 2020), with the most prominent being the graphlet kernel (Shervashidze et al., 2009). Motif-based node embeddings (Dareddy et al., 2019; Rossi et al., 2018) and diffusion operators (Monti et al., 2018; Sankar et al., 2019; Lee et al., 2019) that employ adjacency matrices weighted according to motif occurrences, have recently been proposed for graph representation learning. Our formulation provides a unifying framework for these methods and it is the first to analyse their expressive power. Finally, GNNs that operate in larger induced neighbourhoods (Li et al., 2019; Kim et al., 2019) or higher-order paths (Flam-Shepherd et al., 2020) have prohibitive complexity since the size of these neighbourhoods typically grows exponentially.

## 5 EXPERIMENTAL EVALUATION

In the following section we evaluate GSN in comparison to the state-of-the-art in a variety of datasets from different domains. We are interested in practical scenarios where the collection of subgraphs, as well as their size, are kept small. Depending on the dataset domain we experimented with typical substructure families (*cycles, paths* and *cliques*) and maximum substructure size $k$ (note that for each setting, our substructure collection consists of all the substructures of the family with size $\leq k$). We also experimented with both graphlets and motifs and observed similar performance in most cases. Appendix C provides details and shows additional experiments.

**Synthetic Graph Isomorphism test:** We tested the ability of GSNs to decide if two graphs are non-isomorphic on a collection of Strongly Regular graphs of size up to 35 nodes, attempting to disambiguate pairs with the same number of nodes (for different sizes the problem becomes trivial). As we are only interested in the bias of the architecture itself, we use GSN with random weights to compute graph representations. Two graphs are deemed isomorphic if the Euclidean distance of their representations is smaller than a predefined threshold $\epsilon$. Figure 5 shows the failure percentage of our isomorphism test when using different graphlet substructures (*cycles, paths*, and *cliques*) of varying size $k$. Interestingly, the number of failure cases of GSN decreases rapidly as we increase $k$; cycles and paths of maximum length $k = 6$ are enough to tell apart all the graphs in the dataset. Note that the performance of cliques saturates, possibly because the largest clique in our dataset has 5 nodes. Observe also the discrepancy between GSN-v and GSN-e. In particular, vertex-wise counts do not manage to distinguish all graphs, although missing only a few instances. We hypothesise that this is because edge counts offer a higher level of granularity, allowing GSN to create a finer partition of the nodes in the graph (thus stronger symmetry breaking properties). Finally, 1-WL (Xu et al., 2019) and 2-FWL (Maron et al., 2019a) equivalent models demonstrate 100% failure, as expected from theory.

**TUD Graph Classification Benchmarks:** We evaluate GSN on datasets from the classical TUD benchmarks. We use seven datasets from the domains of bioinformatics and computational social

science and compare against various GNNs and Graph Kernels. Our main GNN baselines are GIN Xu et al. (2019) and PPGNs Maron et al. (2019a). We follow the same evaluation protocol of Xu et al. (2019), performing 10-fold cross-validation and then reporting the performance at the epoch with the best average accuracy across the 10 folds. Table 1 lists all the methods evaluated with the split of Zhang et al. (2018). We select our model by tuning architecture and optimisation hyperparameters and substructure related parameters, that is: (i) $k$, (ii) motifs against graphlets. Following domain evidence we choose the following substructure families: *cycles* for molecules, *cliques* for social networks. Best performing substructures both for GSN-e and GSN-v are reported. As can be seen, our model obtains state-of-the-art performance in most of the datasets, with a considerable margin against the main GNN baselines in some cases.

Table 1: Graph classification accuracy on TUD Dataset. **First**, **Second**, **Third** best methods are highlighted. For GSN, the best performing structure is shown. *Graph Kernel methods.

| Dataset | MUTAG | PTC | Proteins | NCI1 | Collab | IMDB-B | IMDB-M |
|---|---|---|---|---|---|---|---|
| RWK* (Gärtner et al., 2003) | 79.2±2.1 | 55.9±0.3 | 59.6±0.1 | >3 days | N/A | N/A | N/A |
| GK* (k=3) (Shervashidze et al., 2009) | 81.4±1.7 | 55.7±0.5 | 71.4±0.31 | 62.5±0.3 | N/A | N/A | N/A |
| PK* (Neumann et al., 2016) | 76.0±2.7 | 59.5±2.4 | 73.7±0.7 | 82.5±0.5 | N/A | N/A | N/A |
| WL kernel* Shervashidze et al. (2011) | **90.4±5.7** | 59.9±4.3 | 75.0±3.1 | 86.0±1.8 | 78.9±1.9 | 73.8±3.9 | 50.9±3.8 |
| GNTK* (Du et al., 2019a) | 90.0±8.5 | 67.9±6.9 | 75.6±4.2 | 84.2±1.5 | 83.6±1.0 | 76.9±3.6 | 52.8±4.6 |
| DCNN (Atwood & Towsley, 2016) | N/A | N/A | 61.3±1.6 | 56.6±1.0 | 52.1±0.7 | 49.1±1.4 | 33.5±1.4 |
| DGCNN (Zhang et al., 2018) | 85.8±1.8 | 58.6±2.5 | 75.5±0.9 | 74.4±0.5 | 73.8±0.5 | 70.0±0.9 | 47.8±0.9 |
| IGN (Maron et al., 2019b) | 83.9±13.0 | 58.5±6.9 | 76.6±5.5 | 74.3±2.7 | 78.3±2.5 | 72.0±5.5 | 48.7±3.4 |
| GIN (Xu et al., 2019) | 89.4±5.6 | 64.6±7.0 | 76.2±2.8 | 82.7±1.7 | 80.2±1.9 | 75.1±5.1 | 52.3±2.8 |
| PPGNs (Maron et al., 2019a) | 90.6±8.7 | 66.2±6.6 | 77.2±4.7 | 83.2±1.1 | 81.4±1.4 | 73.0±5.8 | 50.5±3.6 |
| Natural GN (de Haan et al., 2020a) | 89.4±1.60 | 66.8±1.79 | 71.7±1.04 | 82.4±1.35 | N/A | 73.5±2.01 | 51.3±1.50 |
| **GSN-e** | 90.6±7.5 | 68.2±7.2 | 76.6±5.0 | 83.5± 2.3 | 85.5±1.2 | 77.8±3.3 | 54.3±3.3 |
| | 6 (cycles) | 6 (cycles) | 4 (cliques) | 15 (cycles) | 3 (triangles) | 5 (cliques) | 5 (cliques) |
| **GSN-v** | 92.2±7.5 | 67.4±5.7 | 74.59±5.0 | 83.5±2.0 | 82.7±1.5 | 76.8±2.0 | 52.6±3.6 |
| | 12 (cycles) | 10 (cycles) | 4 (cliques) | 3 (triangles) | 3 (triangles) | 4 (cliques) | 3 (triangles) |

**ZINC Molecular graphs:** We evaluate GSN on the task of regressing the constrained solubility property of molecules from the ZINC database (Irwin et al., 2012; Kusner et al., 2017; Gómez-Bombarelli et al., 2018; Jin et al., 2018; Dwivedi et al., 2020). Our main baselines are GIN (which we re-implement), as well as a stronger baseline (as in Eq. A.3 and 4 without structural features (see Appendix C.1 for details) that can also take into account edge features (MPNN-*EF*). We then extend both baselines with structural features obtained with $k$-cycle counting (models denoted as GSN and GSN-*EF*) and report the result of the best performing substructure w.r.t. the validation set. The data split into training, validation and test set is obtained from Dwivedi et al. (2020). The evaluation metric and the training loss is the Mean Absolute Error (MAE). Table 2 shows that our model significantly outperforms all the baselines.

In Figure 4, we compare the training and test error for different substructure families (cycles, paths and non-isomorphic trees – for each experiment we use all the substructures of size $\leq k$ in the family). Additionally, we measure the "uniqueness" of the identifiers each substructure yields as follows: for each graph $G$ in the dataset, we measure the number of unique node features $u_G$ (initial node features concatenated with node structural identifiers). Then, we sum them up over the entire training set and divide by the total number of nodes, yielding the disambiguation score $\delta = \frac{\sum_G u_G}{\sum_G |V_G|}$, shown as horizontal bars in Figure 4. A first thing to notice is that the training error is tightly related to the disambiguation score. As identifiers become more discriminative, the model gains expressive power. On the other hand, the test error is not guaranteed to decrease when the identifiers become more discriminative. For example, although cycles have smaller disambiguation scores, they manage to generalise much better than the other substructures, the performance of which is similar to the baseline architecture. This is also observed when comparing against Sato et al. (2020) (-*r* methods in Table 2), where, akin to unique identifiers, random features are used to strengthen the expressivity of GNN architectures. This approach also fails to improve the baseline architectures in terms of mean test error. This validates our intuition that unique identifiers can be hard to generalise when chosen arbitrarily and motivates once more the importance of choosing the identifiers not only based on their discriminative power, but also in a way that allows incorporating the appropriate inductive biases. Finally, we observe a substantial jump in performance when using GSN with cycles of size $k \geq 6$. This is not surprising, as cyclical patterns of such sizes (e.g. aromatic rings) are very common in organic molecules.

Table 2: MAE on *ZINC*. '*-EF*': Additional Edge Features, '*-r*': random features. [†]Our implementation.

| Method | Test MAE |
|---|---|
| GCN | 0.469±0.002 |
| GatedGCN-*EF* | 0.363±0.009 |
| GAT | 0.463±0.002 |
| GIN (Xu et al., 2019) | 0.408±0.008 |
| GIN[†] | 0.288±0.011 |
| GIN[†]-*r* | 0.282±0.021 |
| MPNN-*EF* | 0.184±0.012 |
| MPNN-*EF-r* | 0.193±0.009 |
| **GSN** | 0.139±0.007 |
| **GSN**-*EF* | **0.108**±0.018 |

Table 3: ROC-AUC on `ogbg-molhiv`. '*-AF*': Additional Features, '*-VN*': message passing with a virtual node.

| Method | Training | Validation | Test |
|---|---|---|---|
| GCN-*VN* | 88.65±1.01 | 83.73±0.78 | 74.18±1.22 |
| GCN-*AF* | 88.65±2.19 | 82.04 ±1.41 | 76.06±0.97 |
| GCN-*VN-AF* | 90.07±4.69 | 83.84±0.91 | 75.99±1.19 |
| GIN-*VN* | 93.89±2.96 | 84.10±1.05 | 75.20±1.30 |
| GIN-*AF* | 88.64±2.54 | 82.32±0.90 | 75.58±1.40 |
| GIN-*VN-AF* | 92.73±3.80 | 84.79±0.68 | 77.07±1.49 |
| **GSN**-*VN* | 93.61±1.85 | 84.45±0.97 | 75.88±1.86 |
| **GSN**-*AF* | 88.67±3.26 | 85.17±0.90 | 76.06±1.74 |
| **GSN**-*VN-AF* | **94.30**±3.38 | **86.58**±0.84 | **77.99**±1.00 |

**OGB-MOLHIV:** We use `ogbg-molhiv` from the Open Graph Benchmark (OGB) Hu et al. (2020) as a graph-level binary classification task, where the aim is to predict if a molecule inhibits HIV replication or not. The baseline architecture provided by the authors is a variation of GIN that allows for edge features and is extended with a *virtual node*, *GIN-VN*, or with additional node/edge features, *GIN-AF*, or both, *GIN-VN-AF* (more information in the supplementary material). Similarly to the experiment on ZINC, we extend the baseline settings with cyclical substructure features by treating them in a similar way as node and edge features (*GSN-VN, GSN-AF, GSN-VN-AF*). Using the evaluator provided by the authors, we report the ROC-AUC metric at the epoch with the best validation performance (substructures are also chosen based on the validation set). As shown in Table 3, considerable improvement in the performance of the model in all splits is obtained, thus demonstrating strong generalisation capabilities. Tests with additional chemical features such as formal charge, hybridisation, etc. ('*-AF*' setting in Table 3) show that structure provides important complementary information.

**Structural Features & Message Passing:** We perform an ablation study on the abilities of the structural features to predict the task at the hand, when given as input to a graph-agnostic network. In particular, we compare our best performing GSN with a DeepSets model (Zaheer et al., 2017) that treats the input features and the structural identifiers as a set. For fairness of evaluation the same hyperparameter search is performed for both models (see Appendix C.5). Interestingly, as we show in Table 4, our baseline attains particularly strong performance across a variety of datasets and often outperforms other traditional message passing baselines. This demonstrates the benefits of these additional features and motivates their introduction in GNNs, which are unable to compute them. As expected, we observe that applying message passing on top of these features, brings performance improvements in the vast majority of the cases, sometimes considerably, as in the ZINC dataset.

Table 4: Comparison between DeepSets and GSN with the same structural features

| Dataset | MUTAG | PTC | Proteins | NCI1 | Collab | IMDB-B | IMDB-M | ZINC | MOL-HIV |
|---|---|---|---|---|---|---|---|---|---|
| DeepSets | **93.3**±6.9 | 66.4±6.7 | **77.8**±4.2 | 80.3 ±2.4 | 80.9 ±1.6 | 77.1 ±3.7 | 53.3 ±3.2 | 0.288 ±0.003 | 77.34±1.46 |
| # params | 3K | 2K | 3K | 10K | 30K | 51K | 68K | 366K | 3.4M |
| GSN | 92.8±7.0 | **68.2**±7.2 | **77.8**±5.6 | **83.5**± 2.0 | **85.5**±1.2 | **77.8**±3.3 | **54.3**±3.3 | **0.108** ±0.018 | **77.99**±1.00 |
| # params | 3K | 3K | 3K | 10K | 52K | 65K | 66K | 385K | 3.3M |

## 6 CONCLUSION

In this paper, we propose a novel way to design structure-aware graph neural networks. Motivated by the limitations of traditional GNNs to capture important topological properties of the graph, we formulate a message passing scheme enhanced with structural features that are extracted by counting the appearances of prominent substructures, as domain knowledge suggests. We show both theoretically and empirically that our construction leads to improved expressive power and attains state-of-the-art performance in real-world scenarios. In future work, we will further explore the expressivity of GSNs as an alternative to the $k$-WL tests, as well as their generalisation capabilities. Another important direction is to infer prominent substructures directly from the data and explore the ability of graph neural networks to compose substructures.

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

## A DEFERRED PROOFS

### A.1 GSN IS PERMUTATION EQUIVARIANT

*Proof.* Let $\mathbf{A} \in \mathbb{R}^{n \times n}$ the adjacency matrix of the graph, $\mathbf{H} \in \mathbb{R}^{n \times d_{in}^V}$ the input vertex features, $\mathbf{E} \in \mathbb{R}^{n \times n \times d_{in}^E}$ the input edge features and $\mathbf{E}_i \in \mathbb{R}^{n \times n}$ the edge features at dimension $i$. Let $S_V(\mathbf{A}) \in \mathbb{N}^{n \times d_V}, S_E(\mathbf{A}) \in \mathbb{N}^{n \times n \times d_E}$ the functions generating the vertex and edge structural identifiers respectively. It is obvious that subgraph isomorphism is invariant to the ordering of the vertices, i.e. we will always obtain the same matching between subgraphs $G_S$ and graphs $H$ in the subgraph collection. Thus, each vertex $v$ (edge $(v, u)$) in the graph will be assigned the same structural identifiers $\mathbf{x}_V(v)$ ($\mathbf{x}_E(v, u)$) regarless of the vertex ordering. Thus $S_V$ and $S_E$ are permutation equivariant, i.e. for any permutation matrix $\mathbf{P} \in \{0, 1\}^{n \times n}$ it holds that $S_V(\mathbf{P}\mathbf{A}\mathbf{P}^T) = \mathbf{P}S_V(\mathbf{A})$, $S_E(\mathbf{P}\mathbf{A}\mathbf{P}^T) = \mathbf{P}[S_1^E(\mathbf{A}); \dots; S_{d_E}^E(\mathbf{A})]\mathbf{P}^T$, where the permutation is applied at each slice $S_1^E(\mathbf{A}) \in \mathbb{N}^{n \times n}$ of the tensor $S_E(\mathbf{A})$.

Let $f(\mathbf{A}, \mathbf{H}, \mathbf{E}) \in \mathbb{R}^{n \times d_{out}}$ a GSN layer. We will show that $f$ is permutation equivariant. We need to show that if $\mathbf{Y} = f(\mathbf{A}, \mathbf{X}, \mathbf{E})$ the output of the GSN-layer, then $\mathbf{P}\mathbf{Y} = f(\mathbf{P}\mathbf{A}\mathbf{P}^T, \mathbf{P}\mathbf{X}, \mathbf{P}[\mathbf{E}_1; \dots; \mathbf{E}_{d_E}]\mathbf{P}^T)$ for any permutation matrix $\mathbf{P}$. It is easy to see that GSN-v can can be expressed as a traditional MPNN $g(\mathbf{A}, \mathbf{H}, \mathbf{E}) \in \mathbb{R}^{n \times d_{out}}$ (similar to equations 3 and 4 of the main paper) by replacing the vertex features $\mathbf{H}$ with the concatenation of the input vertex features and the vertex structural identifiers, i.e. $\mathbf{Y} = f(\mathbf{A}, \mathbf{H}, \mathbf{E}) = g(\mathbf{A}, [\mathbf{H}; S_V(\mathbf{A})], \mathbf{E})$. Thus,

$$f(\mathbf{PAP}^T, \mathbf{PH}, \mathbf{PE}_1\mathbf{P}^T, \ldots, \mathbf{PE}_{d_E}\mathbf{P}^T) = g(\mathbf{PAP}^T, [\mathbf{PH}; S_V(\mathbf{PAP}^T)], \mathbf{P}[\mathbf{E}_1, \ldots; \mathbf{E}_{d_E}]\mathbf{P}^T)$$
$$= g(\mathbf{PAP}^T, \mathbf{P}[\mathbf{H}^0; S_V(\mathbf{A})], \mathbf{P}[\mathbf{E}_1; \ldots; \mathbf{E}_{d_E}]\mathbf{P}^T)$$
$$= \mathbf{P}g(\mathbf{A}, [\mathbf{H}^0; S_V(\mathbf{A})], \mathbf{E})$$
$$= \mathbf{PY}$$

where in the last step we used the permutation equivariant property of MPNNs. Similarly, we can show that a GSN-e layer is permutation equivariant, since it can be expressed as a traditional MPNN layer by replacing the edge features with the concatenation of the original edge features and the edge structural identifiers $f(\mathbf{A}, \mathbf{H}, \mathbf{E}) = g(\mathbf{A}, \mathbf{H}, [\mathbf{E}; S_E(A)])$.

Overall, a GSN network is permutation equivariant as composition of permutation equivariant functions, or permutation invariant when composed with a permutation invariant layer at the end, i.e. the READOUT function.

$\square$

## A.2 PROOF OF PROPOSITION 3.1

*Proof.* **GSN is at least as powerful as MPNNs and the 1-WL test:** It is easy to see that GSN model class contains MPNNs and is thus at least as expressive. To show that it is at least as expressive as the 1-WL test, one can repurpose the proof of Theorem 3 in Xu et al. (2019) and demand the injectivity of the update function (w.r.t. both the hidden state $\mathbf{h}^t(v)$ and the message aggregation $\mathrm{MSG}^{t+1}(v)$), and the injectivity of the message aggregation w.r.t. the multiset of the hidden states of the neighbours $\wr \mathbf{h}^t(u) \int_{u \in \mathcal{N}(v)}$. It suffices then to show that if injectivity is preserved then GSNs are at least as powerful as the 1-WL.

We will show the above statement for node-labelled graphs, since traditionally the 1-WL test does not take into account edge labels.[2] We can rephrase the statement as follows: *If GSN deems two graphs $G_1$, $G_2$ as isomorphic, then also 1-WL deems them isomorphic.* Given that the graph-level representation is extracted by a readout function that receives the multiset of the node colours in its input (i.e. the graph-level representation is the node colour histogram at some iteration $t$), then it suffices to show that if for the two graphs the multiset of the node colours that GSN infers is the same, then also 1-WL will infer the same multiset for the two graphs.

Consider the case where the two multisets that GSN extracts are the same: i.e. $\wr \mathbf{h}^t(v) \int_{v \in \mathcal{V}_{G_1}} = \wr \mathbf{h}^t(u) \int_{u \in \mathcal{V}_{G_2}}$. Then both multisets contain the same distinct colour/hidden representations with the exact same multiplicity. Thus, it further suffices to show that if two nodes $v, u$ (that may belong to the same or to different graphs) have the same GSN hidden representations $\mathbf{h}^t(v) = \mathbf{h}^t(u)$ at any iteration $t$, then they will also have the same colours $c^t(v) = c^t(u)$, extracted by 1-WL. Intuitively, this means that GSN creates a partition of the nodes of each graph that is at least as fine-grained as the one created by 1-WL. We prove by induction (similarly to Xu et al. (2019)) that GSN model class contains a model where this holds (w.l.o.g. we show that for GSN-v; same proof applies to GSN-e).

For $t = 0$ the statement holds since the initial node features are the same for both GSN and 1-WL, i.e. $\mathbf{h}^0(v) = c^0(v)$, $\forall v \in \mathcal{V}_{G_1} \cup \mathcal{V}_{G_2}$. Suppose the statement holds for $t - 1$, i.e. $\mathbf{h}^{t-1}(v) = \mathbf{h}^{t-1}(u) \Rightarrow c^{t-1}(v) = c^{t-1}(u)$. Then we show that it also holds for $t$.

Every node hidden representation at step $t$ is updated as follows: $\mathbf{h}^t(v) = \mathrm{UP}^t(\mathbf{h}^{t-1}(v), MSG^t(v))$. Assuming that the update function $\mathrm{UP}^t$ is injective, we have the following: if $\mathbf{h}^t(v) = \mathbf{h}^t(u)$, then:

1. $\mathbf{h}^{t-1}(v) = \mathbf{h}^{t-1}(u)$, which from the induction hypothesis implies that $c^{t-1}(v) = c^{t-1}(u)$.

2. $MSG^t(v) = MSG^t(u)$, where the message function is defined as in Eq. A.3 of the main paper: $MSG^t(v) = \sum_{w \in \mathcal{N}(v)} M^t(\mathbf{h}^{t-1}(v), \mathbf{h}^{t-1}(w), \mathbf{x}_V(v), \mathbf{x}_V(w))$. Additionally here we require

---

[2]if one considers a simple 1-WL extension that concatenates edge labels to neighbour colours, then the same proof applies.

$MSG^t$ to be injective w.r.t. the multiset of the hidden representations of the neighbours. In fact, using Lemma 5 from Xu et al. (2019) we know that there always exists a function $M^t$, such that $MSG^t(v)$ is unique for each multiset $\wr\big(\mathbf{h}^{t-1}(v), \mathbf{h}^{t-1}(w), \mathbf{x}_V(v), \mathbf{x}_V(w)\big)\wr_{w\in\mathcal{N}_v}$, assuming that the domain from where the elements of the multiset originate is countable. Thus,

$$MSG^t(v) = MSG^t(u) \Rightarrow$$
$$\wr\big(\mathbf{h}^{t-1}(v), \mathbf{h}^{t-1}(w), \mathbf{x}_V(v), \mathbf{x}_V(w)\big)\wr_{w\in\mathcal{N}_v} = \wr\big(\mathbf{h}^{t-1}(u), \mathbf{h}^{t-1}(z), \mathbf{x}_V(u), \mathbf{x}_V(z)\big)\wr_{z\in\mathcal{N}_u} \Rightarrow$$
$$\wr\mathbf{h}^{t-1}(w)\wr_{w\in\mathcal{N}_v} = \wr\mathbf{h}^{t-1}(z)\wr_{z\in\mathcal{N}_u}$$

From the induction hypothesis we know that $\mathbf{h}^{t-1}(w) = \mathbf{h}^{t-1}(z)$ implies that $c^{t-1}(w) = c^{t-1}(z)$ for any $w \in \mathcal{N}_v, z \in \mathcal{N}_u$, thus $\wr c^{t-1}(w)\wr_{w\in\mathcal{N}_v} = \wr c^{t-1}(z)\wr_{z\in\mathcal{N}_u}$.

Concluding, given the update rule of 1-WL: $c^t(v) = \text{HASH}\big(c^{t-1}(v), \wr c^{t-1}(w)\wr_{w\in\mathcal{N}_v}\big)$, it holds that $c^t(v) = c^t(u)$.

**When are GSNs strictly more powerful than the 1-WL test?** Arvind et al. (2019) showed that 1-WL, and consequently MPNNs, can count only *forests of stars*. Thus, if the subgraphs are required to be connected, then they can only be star graphs of any size (note that this contains single nodes and single edges). In addition, Chen et al. (2020) showed that 1-WL, and consequently MPNNs, cannot count any connected induced subgraph with 3 or more nodes, i.e. any connected subgraph apart from single nodes and single edges.

Given the first part of the proposition, in order to show that GSNs are strictly more expressive than the 1-WL test, it suffices to show that GSN can distinguish a pair of graphs that 1-WL test deem isomorphic. If $H$ is a substructure that 1-WL cannot learn to count, i.e. the ones mentioned above, then there is at least one pair of graphs with different number of counts of $H$, that 1-WL deems isomorphic. Thus, by assigning counting features to the nodes/edges of the two graphs based on appearances of $H$, a GSN can obtain different representations for $G_1$ and $G_2$ by summing up the features. Hence, $G_1, G_2$ are deemed non-isomorphic. An example is depicted in Figure 2 (left), where the two non-isomorphic graphs are distinguishable by GSN via e.g. cycle counting, but not by 1-WL. $\qquad\square$

### A.3 Proof of Proposition 3.2

Without loss of generality we will show Proposition 3.2 for the case of a single substructure $H$. Moreover, in the context of this proof, for reasons that will be made clear below, we will use directed edge structural identifiers, i.e. edge orbits are defined on directed edges, thus potentially different edge identifiers are assigned to different directions of the same edge. Note that this is not the case for substructures that have a single vertex orbit, such as the ones that we use in our experiments - cliques and cycles. This is because the single vertex orbit induces a single edge orbit as well which is the same both for the directed and the undirected case.

*Proof.* In order to show that GSN-e can express GSN-v, we will first prove that *the vertex identifier of each vertex $v$ can be inferred by the edge identifiers of its incident edges* $\{(v, u) : u \in \mathcal{N}(v)\}$.

We first introduce the following definitions to simplify notation: Recall the definition of a vertex orbit $\text{Orb}^V(v) = \{u \in \mathcal{V}_H : \exists g \in \text{Aut}(H) \text{ s.t. } g(w) = v\}$. For an orbit $\text{Orb}^V(v)$, denote the *orbit neighbourhood* $\mathcal{N}\big(\text{Orb}^V(v)\big)$ as the multiset of the orbits of the neighbours of $v$: $\mathcal{N}\big(\text{Orb}^V(v)\big) = \wr \text{Orb}^V(u) : u \in \mathcal{N}(v)\wr$. Note that the orbit neighbourhood is the same for any vertex $w \in \text{Orb}^V(v)$ and that the neighbourhood is a multiset, since different neighbours might belong to the same orbit. Denote the *orbit degree* $deg(\text{Orb}^V(v))$ as the degree of any vertex in $\text{Orb}^V(v)$ as $deg(v) = deg(\text{Orb}^V(v)) = |\mathcal{N}\big(\text{Orb}^V(v)\big)|$. We will be indexing different vertex orbits as follows: $O_i^V = \text{Orb}^V(v)$.

Regarding the edge orbits, recall that each edge orbit is induced by the orbits of the vertex automorphism group, i.e. $\text{Orb}^E(v, u) = \{(w, z) \in \mathcal{E}_H : \exists g \in \text{Aut}(H) \text{ s.t. } g(w) = v \text{ and } g(z) = u\}$, or al-

ternatively $\mathrm{Orb}^E(v,u) = \{(w,z) \in \mathcal{E}_H : \mathrm{Orb}^V(w) = \mathrm{Orb}^V(v) \text{ and } \mathrm{Orb}^V(z) = \mathrm{Orb}^V(u)\}$. Observe here, that we define edge orbits for directed edges. Let's assume that $\mathrm{Orb}^V(v) = O_i^V$, $\mathrm{Orb}^V(u) = O_j^V$ and $\mathrm{Orb}^E(v,u) = O_{ij}^E$. For ease of notation we will denote $O_{ij}^E = (O_i^V, O_j^V)$.

Recall that vertex structural identifiers are defined as follows (we drop the subscript $H$ for ease of notation):

$$x_i^V(v) = \frac{|\{G_S \simeq H : v \in \mathcal{V}_{G_S} \text{ s.t. } f(v) \in O_i^V\}|}{|\mathrm{Aut}(H)|}, \quad i = 1, \ldots, d_H. \tag{5}$$

and edge structural identifiers for directed edges:

$$x_{ij}^E(u,v) = \frac{|\{G_S \simeq H : (u,v) \in \mathcal{E}_{G_S} \text{ s.t. } (f(u), f(v)) \in O_{ij}^E\}|}{|\mathrm{Aut}(H)|}, \tag{6}$$

Let as assume that there exists only one matched subgraph $G_S \cong H$ and the bijection between $\mathcal{V}_{G_S}$ and $\mathcal{H}$ is denoted as $f$. Then, concerning the vertex orbit $O_i^V$, one of the following holds:

- $v \notin \mathcal{V}_{G_S}$. Then $x_i^V(v) = 0$ and $x_{ij}^E(v,u) = 0$, $\forall u \in \mathcal{N}(v)$,

- $v \in \mathcal{V}_{G_S}$ and $\mathrm{Orb}(f(v)) \neq O_i^V$. Then, $x_i^V(v) = 0$ and $x_{ij}^E(v,u) = 0$, $\forall u \in \mathcal{N}(v)$, otherwise the orbit of $f(v)$ would be $O_i^V$. Note that here the directionality of the edge is important, otherwise an ambiguity would be created and $x_{ij}^E(v,u)$ could have been equal to 1 if the orbit of $f(u)$ is $O_i^V$.

- $v \in \mathcal{V}_{G_S}$ and $\mathrm{Orb}(f(v)) = O_i^V$. Then, $x_i^V(v) = 1$ and since $f(v)$ has exactly $deg(O_i^V)$ neighbours in $H$, then there exist exactly $deg(O_i^V)$ edges incident to $v$ that belong to $G_S$ as well and the set of their edge orbits is exactly $\mathcal{N}(O_i^V)$. In other words, $\sum_{u \in \mathcal{N}(v)} \sum_{j:O_j^V \in \mathcal{N}(O_i^V)} x_{ij}^E(u,v) = deg(O_i^V)$

Thus, by induction, we can easily see that for $m$ matched subgraphs $G_S \cong H$ where $v \in \mathcal{V}_{G_S}$ and $\mathrm{Orb}(f(v)) = O_i^V$, it holds that $x_i^V(v) = m$ and $\sum_{u \in \mathcal{N}(v)} \sum_{j:O_j^V \in \mathcal{N}(O_i^V)} x_{ij}^E(u,v) = m * deg(O_i^V)$. Then it follows that:

$$x_i^V(v) = \frac{1}{deg(O_i^V)} \sum_{u \in \mathcal{N}(v)} \sum_{j:O_j^V \in \mathcal{N}(O_i^V)} x_{ij}^E(u,v) \tag{7}$$

This expression shows that vertex identifiers can be inferred by edge identifiers. The opposite does not necessary hold, since if we know the vertex identifiers $x_i^V(v)$, $x_j^V(u)$ of an edge $(v,u)$ we might not be able to infer $x_{ij}^E(u,v)$. That is because we do not know how many of the subgraphs in $x_i^V(v)$ and $x_j^V(u)$ are common, i.e. they contain both nodes.

The rest of the proof is straightforward: we will assume a GSN-v using substructure counts of the graph $H$, with $L$ layers and width $w$ defined as in the main paper, i.e. at each layer vertex hidden states are updated by $\mathbf{h}^{t+1}(v) = \mathrm{UP}^{t+1}(\mathbf{h}^t(v), \mathrm{MSG}^{t+1}(v))$, where $\mathrm{MSG}^{t+1}(v) = \sum_{u \in \mathcal{N}(v)} M^{t+1}(\mathbf{h}^t(v), \mathbf{h}^t(u), \mathbf{x}_V(v), \mathbf{x}_V(u), \mathbf{e}(\{u,v\}))$. Then, there exists a GSN-e with $L + 1$ layers, where the first layer has width $d_V^{in} + d_V$ and implements the following function:

$\text{UP}^{t+1}\big(\mathbf{h}^t(v), \text{MSG}^{t+1}(v)\big) = [\mathbf{h}^t(v); \text{MSG}^{t+1}(v)]$, where:

$$\text{MSG}^{t+1}(v) = \sum_{u\in\mathcal{N}(v)} M^{t+1}\big(\mathbf{h}^t(v), \mathbf{h}^t(u), \mathbf{x}_E(v,u), \mathbf{e}(\{u,v\})\big)$$

$$= [\frac{1}{deg(O_1^V)} \sum_{u\in\mathcal{N}(v)} \sum_{j:O_j^V\in\mathcal{N}(O_1^V)} x_{ij}^E(u,v); \dots ;$$

$$\frac{1}{deg(O_{d_V}^V)} \sum_{u\in\mathcal{N}(v)} \sum_{j:O_j^V\in\mathcal{N}(O_{d_V}^V)} x_{d_V,j}^E(u,v)]$$

$$= \mathbf{x}_V(v)$$

Note that since $M$ is a universal approximator, then there exist parameters of $M$ with which the above function can be simulated. The next $L$ layers of GSN-e can implement a traditional MPNN where now the input vertex features are $[\mathbf{h}^t(v); \mathbf{x}_V(v)]$ (which is exactly the formulation of GSN-v) and this concludes the proof.

$\square$

# B  COMPARISON WITH HIGHER-ORDER WEISFEILER-LEHMAN TESTS

## B.1  THE WL HIERARCHY

Following the terminology introduced in Maron et al. (2019a), we describe the so-called *Folklore WL family (k-FWL)*. Note that, in the majority of papers on GNN expressivity (Morris et al., 2019; Maron et al., 2019a; Chen et al., 2020) another family of WL tests is discussed, under the terminology *k-WL* with expressive power equal to $(k-1)$-FWL. In contrast, in most graph theory papers on graph isomorphism (Cai et al., 1992; Fürer, 2017; Arvind et al., 2019) the $k$-WL term is used to describe the algorithms referred to as $k$-FWL in GNN papers. Here, we follow the $k$-FWL convention to align with the work mostly related to ours.

The $k$-FWL operates on $k$-tuples of nodes $\mathbf{v} = (v_1, v_2, \dots, v_k)$ to which an initial colour $c^0(\mathbf{v})$ is assigned based on their *isomorphism types* (see section B.3), which can loosely be thought of as a generalisation of isomorphism that also preserves the ordering of the nodes in the tuple. Then, at each iteration the colour is refined as follows:

$$c^{t+1}(\mathbf{v}) = \text{HASH}\Big(c^t(\mathbf{v}), \big\{\!\!\big\{ \big(c^t(\mathbf{v}_{u,1}), c^t(\mathbf{v}_{u,2}), \dots, c^t(\mathbf{v}_{u,k})\big)\big\}\!\!\big\}_{u\in\mathcal{V}}\Big), \tag{8}$$

where $\mathbf{v}_{u,j} = (v_1, v_2, \dots, v_{j-1}, u, v_{j+1}, \dots, v_k)$.

The multiset $\big\{\!\!\big\{ \big(c^t(\mathbf{v}_{u,1}), c^t(\mathbf{v}_{u,2}), \dots, c^t(\mathbf{v}_{u,k})\big)\big\}\!\!\big\}_{u\in\mathcal{V}}$ can be perceived as a form of generalised neighbourhood. Note here, that information is saved in all possible tuples in the graph, thus each $k$-tuple receives information *from the entire graph*, contrary to the *local* nature of the 1-WL test.

## B.2  WHEN CAN GSN BE STRONGER THAN 2-FWL?

In order to give further intuition in the limitations of the 2-FWL, in the following section we describe a specific family of graphs which 2-FWL is known to be unable to disambiguate.

**Definition B.1** (Strongly regular graph)**.** *A* SR$(n,d,\lambda,\mu)$*-graph is a regular graph with $n$ nodes and degree $d$, where every two adjacent vertices have always $\lambda$ mutual neighbours, while every two non-adjacent vertices have always $\mu$ mutual neighbours.*

2-FWL will always decide that a pair of SR graphs with the same parameters $n, d, \lambda, \mu$ are isomorphic, since all 2-tuples of the same isomorphism type have the exact same generalised neighbourhood, leading 2-FWL to converge to its solution at initialisation. On the other hand, as we show in section 5 of the main paper, GSN can tell strongly regular graphs apart by using small-sized substructures. Although it is not clear if there exists a certain substructure collection that results in GSNs that align with the WL hierarchy, we stress that this is not a necessary condition in order to design more powerful GNNs. In particular, the advantages offered by k-WL might not be able to outweigh the

disadvantage of the larger computational complexity introduced. For example, a 2-FWL equivalent GNN will still fail to count 4-cliques (a frequent pattern in social and biological networks) or 8 cycles (a common ring structure in organic molecules). Therefore, we suggest that it might be more appropriate to design powerful GNNs based on the distinct characteristics of the task at hand.

### B.3  WHY DOES 2-FWL FAIL ON STRONGLY REGULAR GRAPHS?

*Proof.* We first rigorously describe what an isomorphism type is. Two k-tuples $\mathbf{v}^a = \{v_1^a, v_2^a, \ldots, v_k^a\}$, $\mathbf{v}^b = \{v_1^b, v_2^b, \ldots, v_k^b\}$ will have the same isomorphism type iff:

- $\forall\, i, j \in \{0, 1, \ldots, k\}, \quad v_i^a = v_j^a \Leftrightarrow v_i^b = v_j^b$

- $\forall\, i, j \in \{0, 1, \ldots, k\}, \quad v_i^a \sim v_j^a \Leftrightarrow v_i^b \sim v_j^b$, where $\sim$ means that the vertices are adjacent.

Note that this is a stronger condition than isomorphism, since the mapping between the vertices of the two tuples needs to preserve order. In case the graph is employed with edge and vertex features, they need to be preserved as well (see Chen et al. (2020)) for the extended case).

For the 2-FWL test, when working with simple undirected graphs without self-loops, we have the following 2-tuple isomorphism types:

- $\mathbf{v} = \{v_1, v_1\}$: *vertex type*. Mapped to the colour $c^{(0)} = c_\alpha$

- $\mathbf{v} = \{v_1, v_2\}$ and $v_1 \nsim v_2$: *non-edge type*. Mapped to the colour $c^{(0)} = c_\beta$

- $\mathbf{v} = \{v_1, v_2\}$ and $v_1 \sim v_2$: *edge type*. Mapped to the colour $c^{(0)} = c_\gamma$

For each 2-tuple $\mathbf{v} = \{v_1, v_2\}$, a generalised "neighbour" is the following tuple: $(\mathbf{v}_{u,1}, \mathbf{v}_{u,2}) = \big((u, v_2), (v_1, u)\big)$, where $u$ is an arbitrary vertex in the graph.

Now, let us consider a strongly regular graph $SR(n,d,\lambda,\mu)$. We have the following cases:

- generalised neighbour of a *vertex type* tuple: $(\mathbf{v}_{u,1}, \mathbf{v}_{u,2}) = \big((u, v_1), (v_1, u)\big)$. The corresponding neighbour colour tuples are:

    - $(c_\alpha, c_\alpha)$ if $v_1 = u$,
    - $(c_\beta, c_\beta)$ if $v_1 \nsim u$,
    - $(c_\gamma, c_\gamma)$ if $v_1 \sim u$.

  The update of the 2-FWL is: $c^{(1)}(\mathbf{v}) = \mathrm{HASH}\Big(c_\alpha, \langle \underbrace{(c_\alpha, c_\alpha)}_{1\ \text{time}}, \underbrace{(c_\beta, c_\beta)}_{n-1-d\ \text{times}}, \underbrace{(c_\gamma, c_\gamma)}_{d\ \text{times}} \rangle\Big)$

  same for all *vertex type* 2-tuples.

- generalised neighbour of a *non-edge type* tuple: $(\mathbf{v}_{u,1}, \mathbf{v}_{u,2}) = \big((u, v_2), (v_1, u)\big)$. The corresponding neighbour colour tuples are:

    - $(c_\alpha, c_\beta)$ if $v_2 = u$,
    - $(c_\beta, c_\alpha)$ if $v_1 = u$,
    - $(c_\gamma, c_\beta)$ if $v_2 \sim u$ and $v_1 \nsim u$,
    - $(c_\beta, c_\gamma)$ if $v_1 \sim u$ and $v_2 \nsim u$,
    - $(c_\beta, c_\beta)$ if $v_1 \nsim u$ and $v_2 \nsim u$,
    - $(c_\gamma, c_\gamma)$ if $v_1 \sim u$ and $v_2 \sim u$.

  The update of the 2-FWL is:

  $c^{(1)}(\mathbf{v}) = \mathrm{HASH}\Big(c_\beta, \langle \underbrace{(c_\alpha, c_\beta)}_{1\ \text{time}}, \underbrace{(c_\beta, c_\alpha)}_{1\ \text{time}}, \underbrace{(c_\gamma, c_\beta)}_{d-\mu\ \text{times}}, \underbrace{(c_\beta, c_\gamma)}_{d-\mu\ \text{times}}, \underbrace{(c_\beta, c_\beta)}_{n-2-(2d-\mu)\ \text{times}}, \underbrace{(c_\gamma, c_\gamma)}_{\mu\ \text{times}} \rangle\Big)$

  same for all *non-edge type* 2-tuples.

- generalised neighbour of an *edge type* tuple:

  - $(c_\alpha, c_\gamma)$ if $v_2 = u$,
  - $(c_\gamma, c_\alpha)$ if $v_1 = u$,
  - $(c_\gamma, c_\beta)$ if $v_2 \sim u$ and $v_1 \not\sim u$,
  - $(c_\beta, c_\gamma)$ if $v_1 \sim u$ and $v_2 \not\sim u$,
  - $(c_\beta, c_\beta)$ if $v_1 \not\sim u$ and $v_2 \not\sim u$,
  - $(c_\gamma, c_\gamma)$ if $v_1 \sim u$ and $v_2 \sim u$.

  The update of the 2-FWL is:

  $$c^{(1)}(\mathbf{v}) = \text{HASH}\Big(c_\gamma, \{\!\{ \underbrace{(c_\alpha, c_\gamma)}_{1\text{ time}}, \underbrace{(c_\gamma, c_\alpha)}_{1\text{ time}}, \underbrace{(c_\gamma, c_\beta)}_{d - \lambda \text{ times}}, \underbrace{(c_\beta, c_\gamma)}_{d - \lambda \text{ times}}, \underbrace{(c_\beta, c_\beta)}_{n - 2 - (2d - \lambda) \text{ times}}, \underbrace{(c_\gamma, c_\gamma)}_{\lambda \text{ times}} \}\!\}\Big)$$

  same for all *edge type* 2-tuples.

From the analysis above, it is clear that all 2-tuples in the graph of the same initial type are assigned the same colour in the 1st iteration of 2-FWL. In other words, the vertices cannot be further partitioned, so the algorithm terminates. Therefore, if two SR graphs have the same parameters $n,d,\lambda,\mu$ then 2-FWL will yield the same colour distribution and thus the graphs will be deemed isomorphic.

□

## C  EXPERIMENTAL SETTINGS - ADDITIONAL DETAILS

In this section, we provide additional implementation details of our experiments. All experiments were performed on a server equipped with 8 Tesla V100 16 GB GPUs, except for the Collab dataset where a Tesla V100 GPU with 32 GB RAM was used due to larger memory requirements (a large percentage of Collab graphs are dense or even nearly complete in some cases). Experimental tracking and hyperparameter optimisation were done via the Weights & Biases platform (wandb) (Biewald, 2020). Our implementation is based on native PyTorch sparse operations (Paszke et al., 2019) in order to ensure complete reproducibility of the results. PyTorch Geometric (Fey & Lenssen, 2019) was used for additional operations (such as preprocessing and data loading).

In each one of the different experiments we aim to show that structural identifiers can be used off-the-shelf and are independent of the architecture. At the same time we aim to suppress the effect of other confounding factors in the model performance, thus wherever possible we build our model on top of a baseline architecture. More details in the relevant subsections. Interestingly, we observed that in most of the cases it was sufficient to replace only the first layer of the baseline architecture with a GSN layer, in order to obtain a boost in performance. This can be understood by considering that if the update and message functions are sufficiently expressive, then they should be able to learn to preserve the input information in their output (in the hidden states of the vertices with regards to vertex counts, or in the hidden states of the endpoints of the edges with regards to edge counts).

Throughout the experimental evaluation the structural identifiers $\mathbf{x}_V(v)$ and $\mathbf{x}_E(\{u, v\})$ are one-hot encoded, by taking into account the unique count values present in the dataset. Other more sophisticated methods can be used, e.g. transformation to continuous features via a normalisation scheme or binning. However, we found that the number of unique values in our datasets were usually relatively small (which is a good indication of recurrent structural roles) and thus such methods were not necessary.

### C.1  GRAPH REGRESSION ON ZINC

**Experimental Details:** The ZINC dataset includes 12k molecular graphs of which 10k form the training set and the remaining 2k are equally split between validation and test (splits obtained from `https://github.com/graphdeeplearning/benchmarking-gnns`). Molecule sizes range from 9 to 37 nodes/atoms. Node features encode the type of atoms and edge features the chemical bonds between them. Again, here node and edge features are one-hot encoded.

We re-implemented the GIN baseline ($GIN^\dagger$ in Table 2 of the main paper). We extended GIN with structural identifiers as in Eq. 11 and 12 (*GSN* model in Table 2). Our stronger baseline (*MPNN-EF* model in Table 2) updates node representations as follows: $\mathbf{h}^{t+1}(v) = \mathrm{UP}^{t+1}(\mathbf{h}^t(v), \mathrm{MSG}^{t+1}(v))$,

$$\mathrm{MSG}^{t+1}(v) = \sum_{u \in \mathcal{N}(v)} M^t\big(\mathbf{h}^t(v), \mathbf{h}^t(u), \mathbf{e}(\{u, v\})\big) \tag{9}$$

$$\mathrm{MSG}^{t+1}(v) = \sum_{u \in \mathcal{N}(v)} M^t\big(\mathbf{h}^t(v), \mathbf{h}^t(u), \mathbf{e}(\{u, v\})\big), \tag{10}$$

where $\mathrm{UP}^t$ and $M^t$ functions are MLPs and $\mathbf{e}(\{u, v\})$ are edge features. Our extension with structural identifiers (*GSN-EF* model in Table 2) is precisely the model of Eq. A.3 or 4 of the main paper. Observe that, probably due to the fact that the ZINC dataset is larger and more stable, the general MPNN-based formulation performs better than the GIN-based counterpart.

Following the same rationale as before, the network configuration is minimally modified w.r.t. the baseline of Dwivedi et al. (2020), while here no hyperparameter tuning is done, since the best performing hyperparameters are provided by the authors. In particular, the parameters are the following: 4 message passing layers, no jumping knowledge, transformation of the output of the last layer by a MLP, readout: sum, batch size: 128, dropout: 0.0, network width: 128, learning rate: 0.001. The learning rate is reduced by 0.5 (decay rate) after 5 epochs (patience) without improvement in the validation loss. Training is stopped when the learning rate reaches the minimum learning rate value of $10^{-5}$. Validation and test metrics are inferred using the model at the last training epoch.

We select our best performing substructure related parameters based on the performance in the validation set in the last epoch. We search cycles with $k = 3, \ldots, 10$, graphlets against motifs, and GSN-v against GSN-e. The chosen hyperparameters for GSN are: *GSN-e, cycle graphlets of 10 nodes* and for GSN-*EF*: *GSN-v, cycle motifs of 8 nodes*. Once the model is chosen, we repeat the experiment 10 times with different seeds and report the mean and standard deviation of the test MAE in the last epoch. This is performed for all 4 models that we implemented and compared ($GIN^\dagger$, *MPNN-EF, GSN, GSN-EF*).

**Fixed parameter budget:** In Table 5 we include an additional comparison between the main methods, where we maintain an approximately constant parameter budget (approx. 100k parameters; we adjust network width to keep this number consistent across different models), as suggested in Dwivedi et al. (2020). Again here, GSN significantly outperforms the baselines, hence the improvement is insensitive to the absolute number of the parameters.

Table 5: MAE on *ZINC*. Experiments with the same parameter budget (100K)

| Method | Test MAE |
|---|---|
| GIN$^\dagger$ | 0.388±0.013 |
| MPNN-*EF* | 0.324±0.006 |
| **GSN** | **0.193±0.010** |
| **GSN-*EF*** | **0.154±0.002** |

**Generalisation: empirical observations:** We repeat the experimental evaluation on ZINC using different fractions of the training set. We compare our best baseline model (MPNN-*EF* in Table 2) against GSN with the best performing substructure (GSN-*EF* in Table 2). In Figure 6, we plot the training and test errors of both methods. Regarding the training error, GSN consistently performs better, following our theoretical analysis on its expressive power. More importantly, GSN manages to generalise much better even with a small fraction of the training dataset. Observe that GSN requires only 20% of the samples to achieve approximately the same test error that MPNN achieves when trained on the entire training set.

**Disambiguation Scores:** In Table 6, we provide the disambiguation scores $\delta$ as defined in section 5 of the main paper for the different types of substructures. Note that these are computed based on vertex structural identifiers (GSN-v).

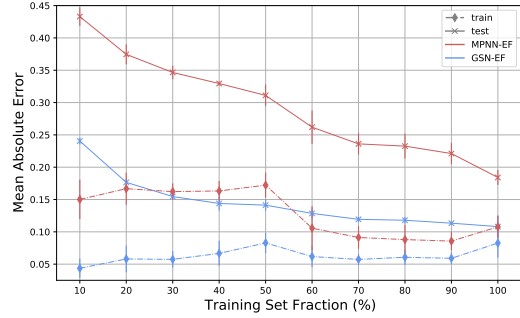

Figure 6: Train (dashed) and test (solid) MAEs for GSN-*EF*(blue) and MPNN-*EF*(red) as a function of the dataset fraction employed in training.

Table 6: Disambiguation scores $\delta$ on *ZINC* for different substructure families and their maximum size $k$. Size $k = 0$ refers to the use of the original node features only.

| k | Cycles | Paths | Trees |
|---|--------|-------|-------|
| 0 | 0.196 | 0.196 | 0.196 |
| 3 | 0.199 | 0.540 | 0.540 |
| 4 | 0.200 | 0.746 | 0.762 |
| 5 | 0.256 | 0.866 | 0.875 |
| 6 | 0.327 | 0.895 | 0.897 |
| 7 | 0.330 | 0.900 | 0.900 |
| 8 | 0.330 | 0.901 | 0.901 |
| 9 | 0.330 | 0.901 | 0.901 |
| 10 | 0.330 | 0.901 | 0.901 |

## C.2 SYNTHETIC EXPERIMENT

For the Strongly Regular graphs dataset (available from `http://users.cecs.anu.edu.au/~bdm/data/graphs.html`) we use all the available families of graphs with size of at most 35 nodes:

- SR(16,6,2,2): 2 graphs
- SR(25,12,5,6): 15 graphs
- SR(26,10,3,4): 10 graphs
- SR(28,12,6,4): 4 graphs
- SR(29,14,6,7): 41 graphs
- SR(35,16,6,8): 3854 graphs
- SR(35,18,9,9): 227 graphs

The total number of non-isomorphic pairs of the same size is $\approx 7 * 10^7$. We used a simple 2-layer architecture with width 64. The message aggregation was performed as in the general formulation of Eq. 5 and 6 of the main paper, where the update and the message functions are MLPs. The prediction is inferred by applying a sum readout function in in the last layer (i.e. a graph-level representation is obtained by summing the node-level representations) and then passing the output through a MLP (we did not use jumping knowledge from intermediate layers (Xu et al., 2018)). Regarding the substructures, we use *graphlet* counting, as certain *motifs* (e.g. cycles of length up to 7) are known to be unable to distinguish strongly regular graphs (since they can be counted by the 2-FWL (Fürer, 2017; Arvind et al., 2019)).

Given the adversities that strongly regular graphs pose in graph isomorphism testing, it would be interesting to see how this method can perform in other categories of hard instances, such as the classical *CFI* counter-examples for k-WL proposed in Cai et al. (1992), and explore further its expressive power and combinatorial properties. We leave this direction to future work.

Table 7: Graph Classification accuracy on various social and biological networks from the TUD Datasets collection `https://chrsmrrs.github.io/datasets/`. Graph Kernel methods are denoted with an *. For completeness we also include methods that were evaluated on potentially different splits. The top three performance scores are highlighted as: **First**, **Second**, **Third**.

| Dataset | MUTAG | PTC | Proteins | NCI1 | Collab | IMDB-B | IMDB-M |
|---|---|---|---|---|---|---|---|
| size | 188 | 344 | 1113 | 4110 | 5000 | 1000 | 1500 |
| classes | 2 | 2 | 2 | 2 | 3 | 2 | 3 |
| avg num. nodes | 17.9 | 25.5 | 39.1 | 29.8 | 74.4 | 19.7 | 13 |
| DGK* (best) (Yanardag & Vishwanathan, 2015) | 87.4 ±2.7 | 60.1±2.6 | 75.7±0.5 | 80.3 ±0.5 | 73.1 ±0.3 | 67.0±0.6 | 44.6±0.5 |
| FSGD* (Verma & Zhang, 2017) | 92.1± | 62.8± | 73.4± | 79.8± | 80.0± | 73.6± | 52.4± |
| AWE-FB*(Ivanov & Burnaev, 2018) | 87.9±9.8 | N/A | N/A | N/A | 71.0±1.5 | 73.1±3.3 | 51.6±4.7 |
| AWE-DD*(Ivanov & Burnaev, 2018) | N/A | N/A | N/A | N/A | 73.9±1.9 | 74.5±5.8 | 51.5±3.6 |
| ECC (Simonovsky & Komodakis, 2017) | 76.1± | N/A | N/A | 76.8± | N/A | N/A | N/A |
| PSCN k=10ᴱ (Niepert et al., 2016) | 92.6±4.2 | 60.0±4.8 | 75.9±2.8 | 78.6±1.9 | 72.6±2.2 | 71.0±2.3 | 45.2±2.8 |
| DiffPool (Ying et al., 2018b) | N/A | N/A | 76.2± | N/A | 75.5 ± | N/A | N/A |
| CCN (Kondor et al., 2018) | 91.6±7.2 | 70.6±7.0 | N/A | 76.3±4.1 | N/A | NA | N/A |
| 1-2-3 GNN (Morris et al., 2019) | 86.1± | 60.9± | 75.5± | 76.2± | N/A | 74.2± | 49.5± |
| RWK* (Gärtner et al., 2003) | 79.2±2.1 | 55.9±0.3 | 59.6±0.1 | ¿3 days | N/A | N/A | N/A |
| GK* (k=3) (Shervashidze et al., 2009) | 81.4±1.7 | 55.7±0.5 | 71.4±0.31 | 62.5±0.3 | N/A | N/A | N/A |
| PK* (Neumann et al., 2016) | 76.0±2.7 | 59.5±2.4 | 73.7±0.7 | 82.5±0.5 | N/A | N/A | N/A |
| WL kernel* (Shervashidze et al., 2011) | **90.4±5.7** | 59.9±4.3 | 75.0±3.1 | **86.0±1.8** | 78.9±1.9 | 73.8±3.9 | 50.9±3.8 |
| GNTK* (Du et al., 2019a) | 90.0±8.5 | **67.9±6.9** | 75.6±4.2 | **84.2±1.5** | **83.6±1.0** | **76.9±3.6** | **52.8±4.6** |
| DCNN (Atwood & Towsley, 2016) | N/A | N/A | 61.3±1.6 | 56.6±1.0 | 52.1±0.7 | 49.1±1.4 | 33.5±1.4 |
| DGCNN (Zhang et al., 2018) | 85.8±1.8 | 58.6±2.5 | 75.5±0.9 | 74.4±0.5 | 73.8±0.5 | 70.0±0.9 | 47.8±0.9 |
| IGN (Maron et al., 2019b) | 83.9±13. | 58.5±6.9 | **76.6±5.5** | 74.3±2.7 | 78.3±2.5 | 72.0±5.5 | 48.7±3.4 |
| GIN (Xu et al., 2019) | 89.4±5.6 | 64.6±7.0 | **76.2±2.8** | 82.7±1.7 | 80.2±1.9 | 75.1±5.1 | 52.3±2.8 |
| PPGNs (Maron et al., 2019a) | **90.6±8.7** | 66.2±6.6 | **77.2±4.7** | 83.2±1.1 | 81.4±1.4 | 73.0±5.8 | 50.5±3.6 |
| Natural GN (de Haan et al., 2020a) | 89.39±1.60 | 66.84±1.79 | 71.71±1.04 | 82.37±1.35 | N/A | 73.50±2.01 | 51.27±1.50 |
| GSN-e (Ours) | **90.6±7.5** | **68.2±7.2** | **76.6±5.0** | 83.5± 2.3 | **85.5±1.2** | **77.8±3.3** | **54.3±3.3** |
|  | 6 (cycles) | 6 (cycles) | 4 (cliques) | 15 (cycles) | 3 (triangles) | 5 (cliques) | 5 (cliques) |
| GSN-v (Ours) | **92.2±7.5** | **67.4±5.7** | 74.6±5.0 | **83.5±2.0** | 82.7±1.5 | **76.8±2.0** | **52.6±3.6** |
|  | 12 (cycles) | 10 (cycles) | 4 (cliques) | 3 (triangles) | 3 (triangles) | 4 (cliques) | 3 (cliques) |

## C.3 TUD Graph Classification Benchmarks

For this family of experiments, due to the usually small size of the datasets, we choose a parameter-efficient architecture, in order to reduce the risk of overfitting. In particular, we follow the simple GIN architecture (Xu et al., 2019) and we concatenate structural identifiers to node or edge features depending on the variant. Then for GSN-v, the hidden representation is updated as follows:

$$\mathbf{h}^{t+1}(v) = \text{UP}^{t+1}\Big(\big[\mathbf{h}^t(v); \mathbf{x}_V(v)\big] + \sum_{u \in \mathcal{N}_v} \big[\mathbf{h}^t(u); \mathbf{x}_V(u)\big]\Big), \tag{11}$$

and for GSN-e:

$$\mathbf{h}^{t+1}(v) = \text{UP}^{t+1}\Big(\big[\mathbf{h}^t(v); \mathbf{x}_E(\{v,v\})\big] + \sum_{u \in \mathcal{N}_v} \big[\mathbf{h}^t(u); \mathbf{x}_E(\{u,v\})\big]\Big), \tag{12}$$

where $\mathbf{x}_E(\{v,v\})$ is a dummy variable (also one-hot encoded) used to distinguish self-loops from edges. Empirically, we did not find training the $\epsilon$ parameter used in GIN to make a difference. Note that this architecture is less expressive than our general formulation. However, we found it to work well in practice for the TUD datasets, possibly due to its simplicity and small number of parameters.

We implement an architecture similar to GIN (Xu et al., 2019), i.e. 4 message passing layers, jumping knowledge from all the layers (Xu et al., 2018) (including the input), transformation of each intermediate graph-level representation by a linear layer, sum readout for biological and mean readout for social networks. Node features are one-hot encodings of the categorical node labels. Similarly to the baseline, the hyperparameters search space is the following: batch size in {32, 128} (except for Collab where only 32 was searched due to GPU memory limits), dropout in {0,0.5}, network width in {16,32} for biological networks, 64 for social networks, learning rate in {0.01, 0.001}, decay rate in {0.5,0.9} and decay steps in {10,50} (number of epochs after which the learning rate is reduced by multiplying with the decay rate). For social networks, since they are not attributed graphs, we also experimented with using the degree as a node feature, but in most cases the structural identifiers were sufficient.

Model selection is done in two stages. First, we choose a substructure that we perceive as promising based on indications from the specific domain: *triangles* for social networks and Proteins, and *6-cycles (motifs)* for molecules. Under this setting we tune model hyperparameters for a GSN-e model. Then, we extend our search to the parameters related to the substructure collection: i.e. the maximum size $k$ and motifs against graphlets. In all the molecular datasets we search cycles with

Table 8: Chosen hyperparameters for each of the two GSN variants for each dataset.

| | Dataset | MUTAG | PTC | Proteins | NCI1 | Collab | IMDB-B | IMDB-M |
|---|---|---|---|---|---|---|---|---|
| | batch size | 32 | 128 | 32 | 32 | 32 | 32 | 32 |
| | width | 32 | 16 | 32 | 32 | 64 | 64 | 64 |
| | decay rate | 0.9 | 0.5 | 0.5 | 0.9 | 0.5 | 0.5 | 0.5 |
| | decay steps | 50 | 50 | 10 | 10 | 50 | 10 | 10 |
| GSN-e | dropout | 0.5 | 0 | 0.5 | 0 | 0 | 0 | 0 |
| | lr | $10^{-3}$ | $10^{-3}$ | $10^{-2}$ | $10^{-3}$ | $10^{-2}$ | $10^{-3}$ | $10^{-3}$ |
| | degree | No | No | No | No | No | No | Yes |
| | substructure type | graphlets | motifs | same | graphlets | same | same | same |
| | substrucure family | cycles | cycles | cliques | cycles | clique | clique | cliques |
| | k | 6 | 6 | 4 | 15 | 3 | 5 | 5 |
| | batch size | 32 | 128 | 32 | 32 | 32 | 32 | 32 |
| | width | 32 | 16 | 32 | 32 | 64 | 64 | 64 |
| | decay rate | 0.9 | 0.5 | 0.5 | 0.9 | 0.5 | 0.5 | 0.5 |
| | decay steps | 50 | 50 | 10 | 10 | 50 | 10 | 10 |
| GSN-v | dropout | 0.5 | 0 | 0.5 | 0 | 0 | 0 | 0 |
| | lr | $10^{-3}$ | $10^{-3}$ | $10^{-2}$ | $10^{-3}$ | $10^{-2}$ | $10^{-3}$ | $10^{-3}$ |
| | degree | No | No | No | No | No | Yes | Yes |
| | substructure type | graphlets | graphlets | same | same | same | same | same |
| | substrucure family | cycles | cycles | cliques | cycles | cliques | clique | cliques |
| | k | 12 | 10 | 4 | 3 | 3 | 4 | 3 |

$k = 3, \ldots, 12$, except for NCI1, where we also consider larger sizes due to the presence of large rings in the dataset (*macrocycles* (Liu et al., 2017)). For social networks, we searched cliques with $k = 3, 4, 5$. In Table 8 we report the hyperparameters chosen by our model selection procedure, including the best performing substructures.

The seven datasets[3] we chose are the intersection of the datasets used by the authors of our main baselines: the Graph Isomorphism Network (GIN) (Xu et al., 2019), a simple, yet powerful GNN with expressive power equal to the 1-WL test, and the Provably Powerful Graph Network (PPGN) (Maron et al., 2019a), a polynomial alternative to the Invariant Graph Network (Maron et al., 2019b), that increases its expressive power to match the 2-FWL. We also compare our results to other GNNs as well as Graph Kernel approaches. Our main baseline from the GK family is the Graph Neural Tangent Kernel (GNTK) (Du et al., 2019a), which is a kernel obtained from a GNN of infinite width. This operates in the Neural Tangent Kernel regime (Jacot et al., 2018; Allen-Zhu et al., 2019; Du et al., 2019b).

Table 7 is an extended version of Table 1 of the main paper, where the most prominent methods are reported, regardless of the splits they were evaluated on. For DGK (best variant) (Yanardag & Vishwanathan, 2015), FSGD (Verma & Zhang, 2017), AWE (Ivanov & Burnaev, 2018), ECC (Simonovsky & Komodakis, 2017), PSCN (Niepert et al., 2016), DiffPool (Ying et al., 2018b), CCN (Kondor et al., 2018) (slightly different setting since they perform a train, validation, test split), 1-2-3 GNN (Morris et al., 2019) and GNTK (Du et al., 2019a), we obtain the results from the original papers. For RWK (Gärtner et al., 2003), GK (Shervashidze et al., 2009), PK (Neumann et al., 2016), DCNN (Atwood & Towsley, 2016) and DGCNN (Zhang et al., 2018), we obtain the results from the DGCNN paper, where the authors reimplemented these methods and evaluated them with the same split. Similarly, we obtain the WLK (Shervashidze et al., 2011) and GIN (Xu et al., 2019) results from the GIN paper, and IGN (Maron et al., 2019b) and PPGN (Maron et al., 2019a) results from the PPGN paper.

### C.4 GRAPH CLASSIFICATION ON `ogbg-molhiv`

The `ogbg-molhiv` dataset contains $\approx$ 41K graphs, with 25.5 nodes and 27.5 edges on average. As most molecular graphs, the average degree is small (2.2) and they exhibit a tree-like structure (average clustering coefficient 0.002). The average diameter is 12 (more details in Hu et al. (2020)).

We follow the design choices of the authors of Hu et al. (2020) and extend their architectures to include structural identifiers. Initial node features and edge features are multi-hot encodings passed through linear layers that project them in the same embedding space, i.e. $\mathbf{h}^0(v) = \mathbf{W}_h^0 \cdot \mathbf{h}_{in}(v)$,

---

[3]more details on the description of the datasets and the corresponding tasks can be found at Xu et al. (2019).

Table 9: Chosen substructures for `ogbg-molhiv`

| Model | GSN-AF | GSN-VN | GSN-VN -AF |
|---|---|---|---|
| features | edges (GSN-e) | vertices (GSN-v) | edges (GSN-e) |
| substructure type | graphlets | graphlets | graphlets |
| substructure family | cycles | cycles | cycles |
| k | 12 | 6 | 6 |

$\mathbf{e}^t(\{v, u\}) = \mathbf{W}_e^t \cdot \mathbf{e}_{in}(\{u, v\})$. The baseline model is a modification of GIN that allows for edge features: for each neighbour, the hidden representation is added to an embedding of its associated edge feature. Then the result is passed through a ReLU non-linearity which produces the neighbour's message. Formally, the aggregation is as follows:

$$\mathbf{h}^{t+1}(v) = \mathrm{UP}^{t+1}\left(\mathbf{h}^t(v) + \sum_{u \in \mathcal{N}_v} \sigma\left(\mathbf{h}^t(u) + \mathbf{e}^t(\{v, u\})\right)\right) \tag{13}$$

A stronger baseline is also proposed by the authors: in order to allow global information to be broadcasted to the nodes, a *virtual node* takes part in the message passing (-*VN* setting in Table 3 of the main paper). The virtual node representation, denoted as $\mathbf{G}^t$, is initialised as a zero vector $\mathbf{G}^0$ and then Message Passing proceeds as follows:

$$\tilde{\mathbf{h}}_v^t = \mathbf{h}^t(v) + \mathbf{G}^t, \ \mathbf{h}^{t+1}(v) = \mathrm{UP}^{t+1}\left(\tilde{\mathbf{h}}^t(v) + \sum_{u \in \mathcal{N}_v} \sigma\left(\tilde{\mathbf{h}}^t(u) + \mathbf{e}^t(\{v, u\})\right)\right),$$

$$\mathbf{G}^{t+1} = MLP^{t+1}\left(\mathbf{G}^t + \sum_{u \in \mathcal{N}_v} \tilde{\mathbf{h}}^t(u)\right) \tag{14}$$

We modify this model, as follows: first the substructure counts are embedded into the same embedding space as the rest of the features. Then, for GSN-v, they are added to the corresponding node embeddings: $\acute{\mathbf{h}}^t(v) = \mathbf{h}^t(v) + \mathbf{W}_V^t \cdot \mathbf{x}_V(v)$, or for GSN-e, they are added to the edge embeddings $\acute{\mathbf{e}}^t(\{v, u\}) = \mathbf{e}^t(\{v, u\}) + \mathbf{W}_E^t \cdot \mathbf{x}_E(\{u, v\})$. Interestingly, even with this simple modification we obtain a considerable improvement in the performance of the model in all splits, thus demonstrating strong generalisation capabilities.

We use the same hyperparameters as the ones provided by the authors, i.e. 5 message passing layers, no jumping knowledge, mean readout, a linear layer applied after the readout, batch size: 32, dropout: 0.5, network width/embedding dimension: 300 (in the ogb implementation the hidden layer of each MLP has dimensions equal to 2*network width, contrary to the rest of the experiments where network width and MLP hidden dimensions are equal), learning rate: 0.001.

We select our best performing substructure related parameters based on the highest validation ROC-AUC (choosing the best scoring epoch as in Hu et al. (2020)). We search cycles with $k = 3, \ldots, 12$, graphlets against motifs, and GSN-v against GSN-e (see Table 9 for the chosen parameters). We repeat the experiment 10 times with different seeds and report the mean and standard deviation of the train, validation and test ROC-AUC, again by choosing the best scoring epoch w.r.t the validation set. We repeat the process for all 3 settings independently (*GSN-VN, GSN-AF, GSN-VN-AF*).

**Impact of the 'Scaffold-based' dataset split:** Motivated by the differences between the train, validation and test distributions in the ogbg-molhiv dataset (the authors of Hu et al. (2020) implement a "scaffold splitting" procedure, i.e. molecules with different 2D-structure are assigned to a different partition of the dataset), we studied how different substructure sizes affect the performance on different sets. In particular, as shown in Table 10, we observed that substructures that yielded the best test score did not always attain best validation performance as well, this confirming the characteristic discrepancy between the test and validation distributions.

Lastly, we note that these results were obtained with a GSN model using the structural features only at its input. In fact, it is possible to 'inject' the structural features at each message passing layer of the architecture, effectively accounting for a skip connection acting on structural identifiers. Although we observe performance improvements, the typical validation/test distribution discrepancy persists. Results are reported in Table 10 ("GSN - skip").

Table 10: ROC-AUC on `ogbg-molhiv`. Performance with different substructures and skip connections for the structural identifiers

| Method | Training | Validation | Test |
|---|---|---|---|
| GSN ($k = 6$) | $94.29 \pm 3.38$ | $\mathbf{86.58} \pm 0.84$ | $77.99 \pm\pm 1.00$ |
| GSN ($k = 8$) | $94.33 \pm 2.38$ | $85.59 \pm 0.82$ | $78.20 \pm 1.69$ |
| GSN - skip ($k = 6$) | $93.77 \pm 3.38$ | $86.44\pm1.09$ | $78.07\pm0.83$ |
| GSN - skip ($k = 8$) | $93.97 \pm 2.70$ | $85.30 \pm 1.01$ | $\mathbf{78.55} \pm 1.25$ |

## C.5 STRUCTURAL FEATURES & MESSAGE PASSING

The baseline architecture treats the input node and edge features, along with the structural identifiers, as a *set*. In particular, we consider each graph as a set of independent edges $(v, u)$ endowed with the features of the endpoint nodes $\mathbf{h}(v), \mathbf{h}(u)$, the structural identifiers $\mathbf{x}_V(v), \mathbf{x}_V(u)$ and the edge features $\mathbf{e}(v, u)$, and we implement a DeepSets universal set function approximator Zaheer et al. (2017) to learn a prediction function $f\bigg( \big\{ \big( \mathbf{h}(v), \mathbf{h}(u), \mathbf{x}_V(v), \mathbf{x}_V(u), \mathbf{e}(v, u) \big) : \{v, u\} \in \mathcal{E}_G \big\} \bigg) =$ $\rho\bigg( \sum_{(v,u)\in\mathcal{E}_G} \phi\big( \mathbf{h}(v), \mathbf{h}(u), \mathbf{x}_V(v), \mathbf{x}_V(u), \mathbf{e}(v, u) \big) \bigg)$, with $\mathcal{E}_G$ the edge set of the graph and $\rho, \phi$, modelled as MLPs. This baseline is naturally extended to the case where we consider edge structural identifiers by replacing $(\mathbf{x}_V(v), \mathbf{x}_V(u))$ with $\mathbf{x}_E(v, u)$. For fairness of evaluation, we follow the exact same parameter tuning procedure as the one we followed for our GSN models for each benchmark, i.e. for the TUD datasets we first tune network and optimisation hyperaparameters (network width was set to be either equal to the ones we tuned for GSN, or such that the absolute number of learnable parameters was equal to those used by GSN; depth of the MLPs was set to 2) and subsequently we choose the substructure related parameters based on the evaluation protocol of Xu et al. (2019). For ZINC and ogbg-molhiv we perform only substructures selection, based on the performance on the validation set. Using the same widths as in GSN leads to smaller baseline models w.r.t the absolute number of parameters, and we interestingly observed this to lead to particularly strong performance in some cases, especially Proteins and MUTAG, where our DeepSets implementation attains state-of-art results. This finding motivated us to explore 'smaller' GSNs (with either reduced layer width or a single message passing layer). These GSN variants exhibited a similar trend, i.e. to perform better than their 'larger' counterparts over these two datasets. We hypothesise this phenomenon to be mostly due to the small size of these datasets, which encourages overfitting when using architectures with larger capacity. In Table 4 in the main paper, we report the result for the best performing architectures, along with the number of learnable parameters.

