# OpenReview forum: "Improving Graph Neural Network Expressivity via Subgraph Isomorphism Counting"
_ICLR.cc/2021/Conference — Reject_

### Official Review · AnonReviewer1 · 2020-10-26
**Official Blind Review #1**

**Rating:** 4
**Confidence:** 4

**Review:**

This work proposes the Graph Substructure Network (GSN) to encode structural roles for different nodes so that the expressivity of Graph Neural Networks is improved. The core idea is to count the number of certain substructures, such as cycles, cliques, and triangles. Then the proposed MPNN encodes such substructure counting information into the message passing. Experimental results show that the proposed method can obtain better performance than the comparing methods.

Strengths:
+ The proposed method can encode important substructure information. It is important for graphs since, in many applications, the substructures can determine the functionality of graphs.

+ The proposed method leads to better performance on different graph classification datasets. The experimental results can show the effectiveness of the proposed method.

Weaknesses:
- The main contribution of this work is counting the substructure information, which can be regarded as the preprocessing of graphs. However, how to properly select the graph set {H_1, … ,H_K}?  With different datasets at hand, the best choice can be quite different. Then how should we apply the proposed method?

- The second concern is the complexity of the proposed methods. Counting different types of substructures can be very time-consuming. As mentioned in this work, the worst case can have O(n^k) complexity. When the graph size is large, we may need to count too many types of substructures.

- The datasets are relatively small; most of them have less than 100 nodes per graph. Then larger datasets, such as RDT-B, RDT-M5K, and RDT-M12K, should be considered. I am wondering how many types of substructures need to be considered for these larger datasets to get better performance.

- This work explicitly encodes substructure information into GNNs. Other existing methods, such as graph pooling methods, which can be considered to implicitly encode structural information. Then advanced pooling methods, such as Diffpool, Structpool, Min-cut Pool, etc., should be discussed and compared.

I am willing to adjust my score if my concerns are properly addressed.

=====Update after rebuttal=====

I have read the authors' rebuttal. Considering the limitations and non-superior performance for larger datasets, I am keeping my score unchanged.

---

> ### Author Response · Authors · 2020-11-23
> **Official response to Reviewer 1**
>
> We thank the reviewer for their constructive feedback and helpful comments.
>
> **1. Substructure selection:**
>
>  We refer the reviewer to our general comment and the discussion in the paper, section 3, paragraph "How to choose the substructures?".
>
> **2. Computational complexity:**
>
> Regarding computational complexity, we refer the reviewer to the general comment. We believe it is important to also stress one additional point: in practice, the number of graphs in the substructure collection does not depend on the graph size, since what we are aiming at is generalisation and not expressivity alone. Even one single substructure family (e.g. cliques or cycles) --or a small subset thereof-- can potentially yield small test error performance if conveying relevant domain knowledge and if it appears frequently within the graph distribution characterising the task at hand. As we highlight in the paper, a small number of small substructures is generally to be preferred aiming at best generalisation.
>
> **3. Larger datasets:**
>
> As suggested by the reviewer, we performed additional experiments on the REDDIT-B and REDDIT-M5K datasets, where the average graph size is larger, comparing a baseline GIN architecture with GSN models equipped either with $3$-cliques or $3$-paths. As mentioned above,  the size of substructures is chosen independently of the size of graphs they are matched onto. We tuned the hyperparameters of these models according to the procedure followed for the TU datasets. Results are reported here below as mean and standard deviation over $10$ random dataset splits:
>
> |            	| REDDIT-B       	| REDDIT-M5K|
> |---|---|---|
> |GIN         	| 0.9005 ± 0.00961   | 0.5695 ± 0.02024||
> |GSN (3-paths)   | 0.9045 ± 0.01507   | 0.5713 ± 0.02398|
> |GSN (3-cliques) | 0.9070 ± 0.01249   | 0.5707 ± 0.02088  |
>
> We observe that GSN improves the performance but only marginally. These results were expected, as we observe that the Reddit datasets mostly contain tree-shaped graphs characterised by an abundance of star graphs and the presence of hub-nodes. Standard message passing may, therefore, be sufficiently expressive. As mentioned in the general comment, experiments on graphs with larger sizes are also performed for the Proteins dataset, where once again substructures of small size are sufficient.
>
>
> **4. Comparison with graph pooling:**
>
> Graph pooling is a component frequently used in modern deep learning architectures for graphs. Although these methods modify the adjacency matrix of the graph, hence its isomorphism class, to date we are not aware of any theoretical evidence that graph pooling improves the expressive power of the GNN, while recently empirical results have also been challenged (Mesquita et al., “Rethinking pooling in graph neural networks”, NeurIPS 2020). In fact, since most pooling methods are formulated as cluster assignment problems, where the clustering matrix is inferred by a traditional MPNN, it is likely that pooling will inherit the limitations of the MPNN. In particular, since the MPNN cannot detect the nodes that belong to a given substructure, then there is no guarantee that these nodes will be clustered together, i.e. there is no guarantee that graph pooling will manage to infer the structural information that we explicitly encode with GSN.
>
> Below we show an example of two non-isomorphic graphs that a cluster assignment-based pooling will fail to distinguish, just as the MPNN that is built on top of:
>
>  - G1 is a disconnected graph with two triangles, G2 is a 6-cycle (Figure 1, Maron et al., "Provably Powerful Graph Networks", NeurIPS 2019).
> - Let the cluster assignment matrix $S = \mathrm{MPNN(G)} \in R^{c \times n}$, where $n$ the number of nodes, $c$ the number of clusters. The coarsened adjacency matrix will be given by $A^\prime = SAS^T$.
> - In this case, the MPNN will assign the same feature $s_u = x \in R^{c \times 1}$ to each node $u$ for both graphs, thus the probability of a node being assigned to a cluster of the coarsened graph will be the same for all the nodes. Then the coarsened adjacency matrix $A^\prime_{ij} = x_i x_j \sum_{u,v \in V_G} A_{uv} = 2|E_G| x_i x_j$. Thus, since both graphs have the same number of edges, then the coarsened graphs will be isomorphic and the neural network will fail to distinguish them.
>
> Overall, although pooling might be implementing mechanisms orthogonal to message passing, there is neither any theoretical evidence that these improve expressivity nor that they can infer the structural information we take into account.

---

### Official Review · AnonReviewer3 · 2020-10-29
**Propose a new method (Graph Substructure Network) for topologically-aware message passing.**

**Rating:** 3
**Confidence:** 5

**Review:**

Reasons for score:
The idea of using small graphs to characterize local topologies and guide message passing in interesting. However, the graph isomorphism computation part has problem. Experimental results are not conclusive. The written need improvement in some part of the manuscript.

Pros: A new method (GSN: Graph Substructure Network) is proposed to a topologically-aware message passing method that better utilize graph substructure information. The method tries to tackles the limitation of traditional GNN in exploring graph structure.  It is a good idea to pass messages differently depending on their local topologies. This is done through using a set of predefine small graphs to characterize local topologies. The authors showed that GSN was more expressive than traditional GNNs. A good number of experimental evaluations were performed.

Cons: It is not clear priorly how to define a good set of small graphs, especially when considering beyond immediate neighbors. In addition, node features are not considered in graph isomorphism, which can lead to incorrect subgraph matching. The experimental results on some of the datasets (such as, MUTAG, PTC, Proteins and NCI1 in table 1) do not appear to be significantly better than those of the previous approaches when considering the variances of different runs. In addition, much better results were reported on the ogb-molhiv leaderboard (https://ogb.stanford.edu/docs/leader_graphprop/#ogbg-molhiv). Figure 1 is confusing.  Should the number in the yellow square on the left be 5? A more self-contained explanation of the figure is appreciated. It will help readers if the authors can visualize a few examples (e.g., contributions of small graphs) to explain why their approach works better.

---

> ### Author Response · Authors · 2020-11-23
> **Official response to Reviewer 3 (2/2)**
>
> **3. Experimental results (2/2)**:
>
> 2. **Regarding the ogbg-molhiv leaderboard:** We respectfully disagree for the following reasons:
>
> - Our intention was to show the our method can easily be adapted in a "plug-and-play" fashion to Graph Neural Network architectures to improve their performance: on this dataset the simple adoption of structural identifiers was sufficient to get ~ 1\% improvement (note that we did not change any of the optimisation or model hyperparameters);
>
> - Some of the methods that achieve better results than ours in the public OGB leaderboard, e.g. GraphNorm, DeeperGCNs, FLAG, deal with aspects that are orthogonal to structural identifiers and GSN and they can seamlessly be combined so as to jointly address different limitations of message passing baselines;
>
> - It is important to stress here that the aforementioned methods are concurrent works and currently undergoing the peer-review process at ICLR, thus we deem the demand for such a comparison unfair.
>
> - The scope of our work is that of studying how the expressivity of Graph Neural Networks can be improved rather than conceiving the most appropriate architecture to model molecular graph distributions. Other methods reported in the leaderboard (HIMP, Morgan fingerprints) are specifically designed to address the latter. We firmly believe that it is more natural and probably more insightful to compare GSN with other architectures focusing on the former (GIN, PPGNN, Natural Graph Networks).
>
> We would also like to point out here an interesting fact regarding the ogbg-molhiv dataset: In order to obtain training, validation and test sets, the authors of the OGB benchmark (Hu et al, NeuIPS 2020), apply a "scaffold splitting" procedure (molecules with different 2D-structure are assigned to a different partition of the dataset). This is a challenging setting for machine learning models since, train, test and validation distributions might differ substantially. We investigated the impact of this aspect on GSN performance and observed that substructures that yielded the best test score did not always attain best validation performance as well, this confirming the characteristic discrepancy between the test and validation distributions. Naturally, coherently with standard model selection procedures, in the main paper we reported the results corresponding to the substructure size with best validation performance, but it is interesting to notice that larger substructures attain better test results with *validation scores which are still the highest reported on this benchmark* (based on the public leaderboard). These results are shown below, and in Section C.4 of the paper Appendix.
>
>
> |k   |Train       	| Validation	| Test |
> |---|---|---|---|
> |6   |94.29 ±  3.38   | 86.58 ±  0.84 | 77.99 ±  1.00|
> |8   |94.33  ±  2.38  | 85.59  ± 0.82 | 78.20  ±  1.69|
>
> One last interesting observation is that these results were obtained with a GSN model using the structural features only at its input. In fact, it is possible to "inject" the structural features at each message passing layer of the architecture, effectively accounting for a skip connection acting on structural identifiers. Further performance improvements --although minor-- were observed, and we report the results here below.
>
> |k  |Train       	| Validation	| Test |
> |---|---|---|---|
> |6  |93.77 ±  3.28   | 86.44 ±  1.09 | 78.07  ±  0.83|
> |8  |93.97  ±  2.70  | 85.30 ±  1.01 | 78.55  ±  1.25|
>
> **4. Figure 1:**
>
> We have checked the computation of the subgraph isomorphism example in Figure 1 and we can confirm its correctness. As already specified in the caption, we consider **induced** paths. Note that the path that is formed by the nodes of the triangle is not induced since its endpoints are connected and, therefore, must not be included in the computation.
>
>
> **5. Contribution of certain substructures:**
>
> An ablation study on different substructures has been  performed on the ZINC dataset, where we show that cycles of certain size achieve significantly better generalisation than others. We contrasted them with paths and trees which, although characterised by much larger disambiguation scores $\delta$ (see also Table 5), do not perform comparably. We refer the reviewer to Figure 4 of the main paper.

---

> ### Author Response · Authors · 2020-11-23
> **Official response to Reviewer 3 (1/2)**
>
> We thank the reviewer for their constructive feedback and helpful comments.
>
> **1. Choice of substructures:**
>
> Regarding the choice of substructures, we invite the reviewer to refer to the general comment above and our discussion in the main paper, Section 3, paragraph "How to choose the substructures?". In particular, let us comment on considering nodes beyond immediate neighbours. Please note that the concept of "neighborhood" is irrelevant to the choice of the substructures, and it is not necessary to constrain the choice to 1-hop neighbourhoods only. This is just a specific case arising for certain substructures (e.g. star graphs, cliques). In fact, other types of substructures have been extensively used throughout our experimentation such as cycles, whose choice has been naturally driven by domain knowledge, while entailing matches with nodes far beyond the 1-hop neighborhood ($[k/2]$ to be specific, where $k$ the length of the cycle).
>
> **2. Inclusion of node features in subgraph matching:**
>
> We remark that the focus of our work is not on the problem of subgraph matching per se, but rather on how this can improve the expressive power of graph neural network architectures. In particular, we deliberately consider only structural matching and refrain from adopting labeled subgraph matching: this last may render harder the selection of a proper substructure bank due to the much larger search space and might compromise the generalisation ability of the architecture due to likely infrequent matches. We respectfully disagree with the reviewer as we believe the presence of node attributes does not compromise the correctness of subgraph matching itself, which is, in our case, structural.
>
> **3. Experimental results (1/2) :**
>
> 1. **Regarding the experimental results on the MUTAG, PTC, Proteins and NCI1 datasets**:  It is a shared concern in the community that some of the TU datasets have certain limitations (e.g. small number of graphs, small average number of vertices) that might render them insufficient for a complete and conclusive evaluation of neural network architectures. Note that the large variances in the performance are also the case for our main baseline architectures (GIN, PPGN) that used the same data split (NB: performance statistics are computed across different dataset splits and not across different initialisations of the architecture). However, we consider the performance gap of GSN to be significant in some TU datasets such as IMDB-B, IMDB-M and especially Collab. Additionally, we remark that we chose to retain the evaluation on these benchmarks in order to compare with a large number of previous approaches which only in a few cases have been tested over more novel and reliable datasets. Either way, it is because we acknowledge the intrinsic issues related to these benchmarks that we extensively evaluated our approach on the novel graph benchmarks ZINC and ogbg-molhiv, where the differences in performance become more obvious and trustworthy.

---

### Official Review · AnonReviewer4 · 2020-10-30
**This paper studies the expressivity of graph neural networks, and proposes a new approach to improve GNN’s expressivity by encoding nodes and edges with features via subgraph isomorphism counting. The proposed solution contains some merit, and the experimental results on graph classification task demonstrates the superiority of the proposed approach.**

**Rating:** 6
**Confidence:** 4

**Review:**

This paper studies the expressivity of graph neural networks, and proposes a new approach to improve GNN’s expressivity by encoding nodes and edges with features via subgraph isomorphism counting. The proposed solution contains some merit, and the experimental results on graph classification task demonstrates the superiority of the proposed approach.

Pros:
1.	This paper addresses an important problem in GNNs, which is to improve the expressivity of GNNs.
2.	The proposed solution is interesting, which is to include how many isomorphic subgraphs from a given list a node or an edge is contained in as additional features.
3.	The experimental results on multiple datasets and different graph-level tasks are better than baselines.
4.	 Nice theoretical analysis.

Cons:
1.	My biggest concern lies in the time complexity of the proposed approach. Although the paper claims that in practice it is not that bad, the worst time complexity is still high. Also, the substructure selection brings us back to feature engineering, or we will face too many possible substructures.
2.	More challenging graph tasks are expected to demonstrate the necessity of the proposed approach.

Detailed comments:
1.	In the abstract and introduction, it is mentioned that existing GNNs are bounded by WL-test, and are not able to detect and count graph structures. It is expected to see experiments are on these more challenging tasks, in addition to graph classification and regression tasks. Graph isomorphism test is an interesting task, and the design is smart.
2.	Discussions on how to select substructures on bigger size of graphs are expected.

---

> ### Author Response · Authors · 2020-11-23
> **Official response to Reviewer 4**
>
> We thank the reviewer for their constructive feedback and helpful comments.
>
> **1a. Computational complexity:**
>
> Regarding computational complexity, we refer the reviewer to the general comment. Here we would like to briefly remark a few points. First, the worst case complexity is rarely encountered in real-world graph distributions, and we empirically show this in Figure 3 of the updated version of the paper. Second, there exist several exact and approximate algorithms that improve on the theoretical worst case complexity.
>
> **1b. Substructure Selection:**
>
> Regarding substructure selection and feature engineering, we refer the reviewer to the general comment. We acknowledge that employing all possible substructures is not feasible and that a selection procedure is necessary. In practice, domain knowledge allowed us to seamlessly narrow down the substructure space and opt for a single substructure family, leaving us with having to choose only the size of the larger substructure $k$. Despite the recent trend of abandoning feature engineering towards end-to-end learning pipelines, we believe that in certain domains this should be embraced with caution, since, so far, the most popular architectures for learning on graphs, i.e. MPNNs, have been shown not to be universal, contrary to CNNs, RNNs, Transformers etc., and in fact they are unable to compute these structural features.
>
> Regarding larger graphs, we stress here that substructure selection should be done independently of the size of the graph in order to obtain better generalisation properties. Please refer to the general comment and our paper for further discussion.
>
>
> **2. More challenging graph tasks:**
>
> In this paper, we study expressivity from the perspective of graph isomorphism and thus we performed experiments on this task in order to validate our theoretical claims.  However, we certainly agree with the reviewer that dealing with more challenging (computationally hard in the worst case) graph tasks (e.g. subgraph counting, graph edit distance, combinatorial optimisation) is an interesting topic not only experimentally, but also and foremost from a theoretical standpoint (e.g. can we combine message passing with small substructures to compose larger ones?, can we improve size generalisation using graph substructures? - see "On Size Generalisation in Graph Neural Networks", Anonymous, submitted to ICLR 2021) Thus,  experimentation on such tasks should be accompanied by the appropriate theoretical analysis. These questions were beyond the scope of this paper and we are planning to address them in future work.

---

### Official Review · AnonReviewer2 · 2020-11-02
**A natural idea to increase the expressive power of GNNs that would benefit from more theoretical results and more experimental evaluation**

**Rating:** 5
**Confidence:** 5

**Review:**

This paper presents a natural extension of Message Passing Neural Net (MPNN)  by incorporating structural features. These structural features are computed as the counts from different substructures (like small lines, stars or complete graphs) induced in the original graph. These counts are combined to obtain a new feature per node or per edge. Then these features are used in a standard MPNN. The authors then show that the resulting GNN is more expressive and they validate this claim experimentally.

This idea is interesting and clearly explained in the paper but I think the paper could be greatly improved after addressing the following issues:

1- the authors should clarify their position with regards to invariance. Indeed, as explained shortly on page 3 when commenting Loukas(2020), it is easy to make a GNN powerful if we remove the constraint to be invariant (or equivariant). Hence, as I understand it, the authors are proposing an algorithm that is equivariant. If this is the case, it would be great to have a clear formal statement that GSN are equivariant and to give a mathematical proof.

2- the theoretical content of the paper should be improved:

 a- the first part of Proposition 3.1 is straightforward and I find the wording of the second statement unclear: what does '... or any not necessarily induced subgraph...' mean?

 b- from a theoretical perspective, it seems that both GSN-v abd GSN-e have the same expressive power. Is it true?

 c- a similar idea as the one presented in this paper was presented in : Coloring graph neural networks for node disambiguation  by George Dasoulas, Ludovic Dos Santos, Kevin Scaman, Aladin Virmaux https://arxiv.org/abs/1912.06058 [arxiv-col] The main advantage of the current paper as opposed to [arxiv-col] is to propose an explicit coloring thanks to the structural features. But the theoretical analysis made in [arxiv-col]  goes much deeper than this paper and probably could be adapted by the authors. For example, Corollary 3.1 could probably be replaced by a universality property, i.e. GSN with k=n-1 is universal.

3- the experimental evaluation is not convincing. To make it more convincing, the authors should include an ablation study for all their experiments by comparing their performances with the performances obtained with the structural features only. Such an ablation study would show the benefit of adding the MPNN on top of these features.

[After rebuttal] I think the authors improved their paper by taking into account the remarks. Given the last results obtained in Table 4, it looks like the structural features are indeed very good features in practice as they allow to boost the performances of a very simple invariant architecture like Deepset. I think the authors should explore how they can combine this approach with the coloring approach to get better GNN.

---

> ### Author Response · Authors · 2020-11-23
> **Official response to Reviewer 2 (2/2)**
>
> **2.c. Relation to Dasoulas et al., IJCAI 2020**
>
> We thank the reviewer for pointing out this relevant paper. Indeed, enhancing the node features with different colourings leads to a unique identification scheme similar to Loukas, ICLR 2020, which was an important inspiration for our work. In fact, an interesting observation than can be made is that substructures and colourings can be used in combination, in order to benefit from the best of both worlds. In particular, GSN is permutation equivariant, but universality is so far guaranteed when the size of the substructures is in the order of the size of the graph $k=n-1$ (subject to the validity of the reconstruction conjecture).  On the other hand, colourings and in general arbitrarily chosen unique identifiers are universal, but compromise the permutation equivariance property of GNNs, while in order for a colouring graph network to be equivariant, all possible colour combinations should be used (equation (6) of the mentioned paper: there exist $\prod_{k=1}^K |V_k|!$ different colourings, where each $V_k$ contains nodes with the identical node attributes).  However, by counting substructures, we can reduce the size of the sets $V_k$, since structural identifiers usually manage to break a significant amount of symmetries in the graph (e.g. see Table 5). Thus, by combining the two approaches, one can obtain universality and permutation equivariance while using fewer substructures and less colourings at the same time. This can be an interesting direction for future work.
>
> Regarding Corollary 3.1, indeed being able to solve graph isomorphism implies universality as shown in Dasoulas et al., IJCAI 2020 and Chen et al., NeurIPS 2019. The corollary has been rephrased in the updated version of the paper.
>
>
>
> **3. Comparison with structural features only:**
>
> We thank the reviewer for suggesting this interesting ablation study. Agreeing on the relevance of studying the benefit of message passing of structural identifiers, we comprehensively tested a connectivity-agnostic baseline (similar baselines have been used in Dwivedi et al., arXiv 2020) treating input node and edge features, along with the structural identifiers, as a *set*. In particular, we consider each graph as a set of independent edges $(v,u)$ endowed with the features of the endpoint nodes, their structural identifiers and edge features and we implement a DeepSets universal set function approximator (Zaheer et al., NIPS 2017) to learn a prediction function $f\bigg(\\{\big(h(v), h(u), x_V(v), x_V(u), e(v,u)\big) : \\{v,u\\}\in E_G\\}\bigg) = \rho\bigg(\sum_{(v,u)\in E_G} \phi\big(h(v), h(u), x_V(v), x_V(u), e(v,u)\big)\bigg)$, with $E_G$ the edge set of the graph and $\rho, \phi$, modeled as MLPs. This baseline is naturally extended to the case where we consider edge structural identifiers by replacing $x_V(v), x_V(u)$ with $x_E(v,u)$. For fairness of evaluation, we follow the exact same hyperparameter tuning procedure as the one we followed for our GSN models for each benchmark. We refer the reviewer and interested reader to the Appendix C.5 in the updated version of the paper for additional experimental details.
>
> The results are reported in Table 4 of the main paper.  As can be clearly seen, our baseline attains particularly strong performance across a variety of datasets and often outperforms other traditional message passing baselines which do not make use of structural identifiers. This demonstrates once more the benefits of these additional features and motivates their introduction in GNNs, which are unable to compute them. As expected, we observe the additional message passing layers applied on top of these features by GSN to generally bring performance improvements, sometimes considerably, as in the ZINC dataset. Interestingly, our DeepSets baseline performs on par with GSN on the Proteins benchmark, and better than any other model --including GSN-- on MUTAG, where it attains state-of-the-art performance. This last is a relatively small dataset ($188$ graphs) and we hypothesize our baseline is able to effectively balance model capacity and inductive priors into a strong sample-efficient architecture.

---

> ### Author Response · Authors · 2020-11-23
> **Official response to Reviewer 2 (1/2)**
>
> We thank the reviewer for their insightful comments regarding the theoretical content of the paper. In particular, we found the observation regarding the relation between the GSN-e and GSN-v to be particularly interesting, and this led us to a new finding that deepened our theoretical analysis, i.e. we showed that GSN-e is at least as powerful as GSN-v. Their comments allowed us to improve the paper regarding the exposition and our theoretical claims. We addressed them as follows:
>
>
> **1. Equivariance of GSN**
> As the reviewer correctly points out, GNNs can become universal by dropping the permutation equivariant property. However, given that the absence of this inductive bias usually compromises generalisation, our goal is to design a GNN that is provably more expressive, while at the same time permutation equivariant. It is easy to see that this property holds for GSN, since the process that generates the structural identifiers is permutation equivariant. In the updated version of the paper, Appendix, Section A.1, we provide a short proof of GSN's permutation equivariance property.
>
> **2a. Clarifications on Proposition 3.1**
> Let us clarify our statement "is strictly more powerful than MPNN and the 1-WL test when $H$ is any induced subgraph except for single edges and single nodes, or any not necessarily induced subgraph except from star graphs of any size.": As mentioned in the preliminaries, we make the distinction between "subgraphs" and "induced subgraphs". In the first case, the edges of the subgraph $G_S \cong H$ ($H$ is the graph that we are seeking to detect in the graph $G$) need to be a subset of the edges of the graph $G$: $E_{G_S} \subseteq E_G$. In the second case, if two nodes $u,v$ are connected in the original graph $G$ then they must be connected also in the subgraph $G_S$, i.e. $E_{G_S} = E_G \cap \big(V_{G_S} \times V_{G_S}\big)$. Arvind et al. showed that for the case of "subgraphs", 1-WL can count only star graphs of any size (including single nodes and edges). This is what we refer to when we mention "not necessarily induced subgraphs" and the phrasing was done in this way in order to emphasise the difference with "induced subgraphs". For the case of "induced subgraphs", Arvind et al. showed that 1-WL cannot count any subgraph but single nodes and edges. We rephrased proposition 3.1 in order to make it more clear. The first part of the proposition which was straightforward was removed.
>
> **2.b Do GSN-e and GSN-v have the same expressive power**
> We thank the reviewer for bringing our attention to this very interesting question. We theoretically analysed the expressive power of the two variants and it turns out that **GSN-e is at least as expressive as GSN-v**.  To prove this claim, we are going to show that for each substructure $H$ and any GSN-v that uses counts of $H$, there exists a GSN-e that can "simulate" the GSN-v. That is because for each graph $H$ in the collection, the vertex structural identifiers can be reconstructed by the corresponding edge identifiers. We formalise this statement in proposition 3.2 in the updated version of the paper and we invite the reviewer and the interested reader to take a deeper look to the complete proof in the paper Appendix, Section A.3.
>
> Interestingly, when using GSN with random weights to test graph isomorphism in Strongly Regular graphs (see Figure 4), we can see that for certain substructures $H$, GSN-e has less failure cases than GSN-v, thus we suspect that there might exist graphs $G$ that GSN-v does not have the ability to distinguish, while GSN-e does.

---

### Author Response · Authors · 2020-11-23
**Summary of updates in the revised version of the paper**

We thank the reviewers for their detailed feedback and insightful comments. In these two posts, we summarise the updates made in the paper during the rebuttal period and respond to common concerns raised by the reviewers. We also provide a separate response to each reviewer in order to address their comments individually.

### **Paper updates**:

We updated our paper as follows:
1. **Theoretical analysis**:

     - Exposition: we improved the exposition of some of our claims, as advised by R2, in order to make them more clear.
     - Permutation equivariance: we clarified that GSN is indeed permutation equivariant and provided a short proof in the supplementary material.
     - Expressive power of GSN-e vs GSN-v: we theoretically analysed the two variants and found that **GSN-e is at least as expressive as GSN-v**. We provide the proof in the supplementary material, showing that edge structural identifiers can be used to infer vertex structural identifiers.

2. **Comparison with structural features only**: We conducted an extensive experimental evaluation on all the datasets that we used in the paper, where we discarded message passing and  trained a DeepSets network that receives at its input the structural features only (as well as node and edge features for fairness of evaluation). Interestingly, we find that this architecture is a strong baseline across many datasets, and that as expected, when used in synergy with message passing the performance generally improves. With this experiment we show the benefit of message passing, but also the importance of the relevant structural inductive biases for each task.

3. **Time complexity**: We include visualisations of the empirical runtime of the subgraph isomorphism counting preprocessing stage of our algorithm on three representative datasets (taken from the domains of social science, bioinformatics and chemistry). With that, we aim to show that the expected complexity is usually significantly better than the worst case, making our algorithm practical and easy-to-use.

---

> ### Author Response · Authors · 2020-11-23
> **Time complexity**
>
> ### **Time complexity**:
>
> R4 and R1 raised concerns about the time complexity of the approach. We acknowledge the concern about the theoretical worst-case time complexity for the precomputation of the structural identifiers needed by the GSN models, which occurs when the underlying graph is fully connected. However, we would like to respectfully remark the fact that in most real-world applications of interest it is unlikely to encounter the worst-case scenario, due to the typically sparse nature of the graphs. In order to empirically validate this intuition, we performed timing experiments over three datasets (IMDB-MULTI, Proteins and ZINC), which are representative of the three main application domains considered in our work (social science, bioinformatics, and chemistry, respectively). The results have been reported in the main paper, Figure 3. In this experiment we measure the time required for the computation of structural identifiers (cliques for IMDB-MULTI and Proteins, and cycles for ZINC) of various sizes $k$ as a function of graph size $n$ (number of nodes). We compare the time required on these two benchmarks with those required on a control dataset consisting of complete graphs with different sizes, which represent valid proxies for a worst-case subgraph isomorphism computation ($k log(n)$ growth in logarithmic scale). From the results reported in Figure 3, we make the following observations.
>
> -  In all the three benchmarks the computational cost is substantially lower than the worst case, and in ZINC and Proteins, the runtime scales dramatically better.
> -  The three benchmarks significantly differ in the observed growth trend behavior, with ZINC and Proteins requiring much lighter computation. We attribute this to the sparsity of most graphs in these datasets, in contrast with the presence of strongly tight (often completely connected) communities in the IMDB-MULTI dataset.
>
> Finally, we refer the reviewers to the discussion regarding computational complexity present in our main paper (Section 3, paragraph "Complexity"), where we also make reference to algorithms in the literature that propose exact and approximate algorithmic approaches to also improve on the worst-case theoretical complexity.

---

> ### Author Response · Authors · 2020-11-23
> **Substructure Selection**
>
> ### **Substructure Selection**:
>
> R1, R3 and R4 raised concerns about the substructure selection procedure. We would like to refer the reviewers and the interested reader to the dedicated discussion on the issue in Section 3 of the main paper ("How to choose the substructures?"). In particular, we would like to stress that in practice, the size of the substructures that need to be chosen is **independent of the size of the graph**, i.e. $k = \mathcal{O}(1)$ and this is mainly for two reasons:
>
> - Expressivity - substructures as a symmetry breaking mechanism: As we show in Table 5 in the appendix, substructure counts can be used in order to disambiguate nodes, i.e. in order to provide nodes with different identifiers. In particular, it is shown that certain substructures of small size ($\approx$ 5) can disambiguate a large percentage of the nodes and thus boost the expressivity of the GNN by a considerable margin. This is also experimentally shown in Figure 3, where the more discriminatory the substructures are, the more the training error decreases. While in theory, given the reconstruction conjecture, universality is achieved with $k = n - 1$ sized substructures, the graphs requiring such large substructures are unlikely to be encountered in practice, since they are mostly represented by specific counter-examples that are usually even hard to construct (e.g. see Cai, Fürer, Immerman, Combinatorica, 1992).
>
> - Generalisation - substructures as inductive biases: As we show in Figure 4 in the main paper, when choosing large and diverse substructures, the GNN becomes more expressive due to their greater discriminative power, but this does not provide any guarantees on generalisation. Thus, the substructures need to be selected in a way that provides the appropriate inductive biases with respect to the task at hand. Tasks we usually deal with in real-world settings depend on the local properties of graphs, this incentivising us to choose substructures with a small size, independent of that of the network (e.g although in Proteins we are dealing with graphs of size up to 600 nodes, triangles and 4-cliques are sufficient to obtain improved performance).
>
>
> We acknowledge that a relevant research direction is the construction of an algorithmic procedure to automatically select the substructures. However, this was beyond the scope of our work, where our main goal was to devise a general framework based on substructures that provably improves expressivity. We thus envision to explore this direction as future work.
>
> As a reminder, let us mention here that in this work substructure selection was straightforward: we used a single substructure family for each dataset  (cliques, cycles, etc.), based on what domain knowledge suggests regarding the characteristics of the underlying graph distribution. This narrowed down the substructure search space significantly, as we only had to tune one parameter, i.e. the size of the largest substructure $k$.

---

### Decision · Program_Chairs · 2021-01-07
**Final Decision**

**Decision:**

Reject

**Comment:**

We thank the authors for their detailed responses and the revised version, which addresses several of the questions raised by the reviewers.

The paper is correct and clearly written. All reviewers agree that the idea to add structural features in the message passing of graph neural networks is sensible. While different from previous work, the novelty is a bit incremental though, particularly given the previous work on colored graph neural network. The significance of the work is weak, given 1) the need to select "by hand" structural features that are passed as information, 2) the increased time complexity to compute the structural features compared to other GCNN, and 3) the experimental results that suggest that the benefit of the new approach is limited, particularly on challenging task.

To summarize, this is not a bad paper, but we consider it below the standard of ICLR in terms of originality and significance.